# Antimicrobial Resistance and Mechanisms of Azithromycin Resistance in Nontyphoidal *Salmonella* Isolates in Taiwan, 2017 to 2018

Chien-Shun Chiou,[a] Yu-Ping Hong,[a] You-Wun Wang,[a] Bo-Han Chen,[a] Ru-Hsiou Teng,[a] Hui-Yung Song,[a] Ying-Shu Liao[a]

aCenter for Diagnostics and Vaccine Development, Centers for Disease Control, Taipei, Taiwan

**ABSTRACT** Antimicrobial resistance was investigated in 2,341 nontyphoidal *Salmonella* (NTS) isolates recovered from humans in Taiwan from 2017 to 2018 using antimicrobial susceptibility testing. Azithromycin resistance determinants were detected in 175 selected isolates using PCR and confirmed in 81 selected isolates using whole-genome sequencing. Multidrug resistance was found in 47.3% of total isolates and 96.2% of *Salmonella enterica* serovar Anatum and 81.7% of *S. enterica* serovar Typhimurium isolates. Resistance to the conventional first-line drugs (ampicillin, chloramphenicol, and cotrimoxazole), cefotaxime and ceftazidime, and ciprofloxacin was found in 32.5 to 49.0%, 20.3 to 20.4%, and 3.2% of isolates, respectively. A total of 76 (3.1%) isolates were resistant to azithromycin, which was associated with *mph*(A), *erm*(42), *erm*(B), and possibly the enhanced expression of efflux pump(s) due to *ramAp* or defective *ramR*. *mph*(A) was found in 53% of the 76 azithromycin-resistant isolates from 11 serovars and located in an IS*26*-*mph*(A)-*mrx*(A)-*mphR*(A)-IS*6100* unit in various incompatibility plasmids and the chromosomes. *erm*(42) in *S. enterica* serovar Albany was carried by an integrative and conjugative element, ICE_erm42, and in *S. enterica* serovar Enteritidis and *S. Typhi*murium was located in IS*26* composite transposons in the chromosomes. *erm*(B) was carried by IncI1-I($\alpha$) plasmids in *S. Enteritidis* and *S. Typhi*murium. *ramAp* was a plasmid-borne *ramA*, a regulatory activator of efflux pump(s), found in only *S. enterica* serovar Goldcoast. Since the azithromycin resistance determinants are primarily carried on mobile genetic elements, they could easily be disseminated among human bacterial pathogens. The *ramAp*-carrying *S.* Goldcoast isolates displayed azithromycin MICs of 16 to 32 mg/L. Thus, the epidemiological cutoff value of ≤16 mg/L of azithromycin proposed for wild-type NTS should be reconsidered.

**IMPORTANCE** Antimicrobial resistance in NTS isolates is a major public health concern in Taiwan, and the mechanisms of azithromycin resistance are rarely investigated. Azithromycin and carbapenems are the last resort for the treatment of invasive salmonellosis caused by multidrug-resistant (MDR) and extensively drug-resistant *Salmonella* strains. Our study reports the epidemiological trend of resistance in NTS in Taiwan and the genetic determinants involved in azithromycin resistance. We point out that nearly half of NTS isolates from 2017 to 2018 are MDR, and 20% are resistant to third-generation cephalosporins. The azithromycin resistance rate (3.1%) for the NTS isolates from Taiwan is much higher than those for the NTS isolates from the United States and Europe. Our study also indicates that azithromycin resistance is primarily mediated by *mph*(A), *erm*(42), *erm* (B), and *ramAp*, which are frequently carried on mobile genetic elements. Thus, the azithromycin resistance determinants could be expected to be disseminated among diverse bacterial pathogens.

**KEYWORDS** *Salmonella*, nontyphoidal *Salmonella* (NTS), antimicrobial resistance, azithromycin, mechanisms of azithromycin resistance

Address correspondence to Chien-Shun Chiou, nipmcsc@cdc.gov.tw.

The authors declare no conflict of interest.

The *Salmonella* genus comprises two species, *S. enterica* and *S. bongori*, and more than 2,600 serovars (1), which can be grouped into typhoidal and nontyphoidal *Salmonella* (NTS) serovars (2). The typhoidal *Salmonella* serovar, *S.* Typhi, and paratyphoidal serovars, *S.* Paratyphi A, *S.* Paratyphi B, *S.* Paratyphi C, and *S.* Sendai, can cause invasive systemic infections in humans and higher primates, resulting in an estimated 21,650,974 cases of typhoid fever and 216,510 deaths, along with 5,412,744 cases of paratyphoid fever, globally in 2000 (3). NTS is estimated to cause 93.8 million illnesses, of which 80.3 million are foodborne, and 155,000 deaths each year (4). Invasive NTS disease is a major cause of global morbidity and mortality, with the highest incidence in sub-Saharan Africa (5). NTS is estimated to cause 535,000 cases of invasive disease, with 77,500 deaths globally in 2017 (5). Among foodborne pathogens, NTS is the second leading cause of illness and the largest cause of hospitalization and death in the United States (6).

NTS infections in healthy humans usually result in only a mild and self-limiting symptomatic illness. Antimicrobial therapy can prolong the duration of excretion of NTS, and there is no evidence of benefit for antimicrobials in NTS diarrhea in healthy people; therefore, antimicrobial therapy is only recommended for people with severe illness, invasive disease, or certain risk groups, including infants, the elderly, and immunocompromised individuals (7, 8). Ampicillin, chloramphenicol, and cotrimoxazole used to be the first-line antimicrobials to treat salmonellosis (9). Due to the widespread resistance of *Salmonella* serovars to conventional first-line drugs, fluoroquinolones (e.g., ciprofloxacin), third-generation cephalosporins (e.g., ceftriaxone), macrolides (e.g., azithromycin), and carbapenems (e.g., meropenem) have been indicated as the critically important antimicrobials for the treatment of salmonellosis (9, 10). Thus, surveillance of the resistance to the critically important drugs in *Salmonella* isolates is of great medical concern. In the United States, resistance in human NTS isolates from 2014 was low to ceftriaxone (2.4%) and very rare to ciprofloxacin (0.4%) and azithromycin (<0.1%) (11). In Europe, resistance in human NTS isolates from 2020 was 14.1% for ciprofloxacin, 0.8% for cefotaxime and ceftazidime, and 0.8% for azithromycin (12).

Azithromycin, a semisynthetic macrolide, has been widely used to treat a variety of bacterial infections, including invasive salmonellosis (13). This broad-spectrum agent had been used in massive treatments to eradicate trachoma and reduce all-cause mortality in children (14). However, massive use of azithromycin selects for resistance to this antimicrobial in bacteria (15). Although azithromycin resistance in NTS is rare, resistance has been increasing over time and seems to be more prevalent in strains with multidrug resistance and fluoroquinolone resistance (16, 17).

Macrolides, such as azithromycin and erythromycin, inhibit bacteria by binding to bacterial 50S ribosomal subunits to hinder mRNA translation (13). Bacteria can develop resistance to macrolides, including azithromycin, through target alterations in 23S rRNA and ribosomal proteins L4 and L22, methylation of 23S rRNA by methyltransferases, decreased uptake of drugs via increased extrusion by efflux pumps and decreased permeability of the outer membrane, and inactivation of drugs by modifying enzymes (13). Several mechanisms of azithromycin resistance have been found in *Salmonella*, including modification of the drug by Mph(A), a macrolide 2′-phosphotransferase (18), methylation of 23S rRNA by ErmB and Erm42, rRNA adenine N-6-methyltransferases (19, 20), and increased drug extrusion by an AcrAB-TolC efflux pump that has an R717 mutation in AcrB (21). The resistance determinants can be carried by plasmids (22), transposons, including integrative and conjugative elements (23), and chromosomes (24).

Salmonellosis caused by NTS is common in Taiwan, though the number of cases has not been well estimated. Several studies have indicated that NTS isolates from Taiwan were highly resistant to antimicrobials. Lauderdale et al. (25) reported that 41.0% of 798 isolates recovered from humans in Taiwan in 1998 to 2002 were resistant to ampicillin, chloramphenicol, streptomycin, sulfisoxazole, and tetracycline, and 27.9% were nonsusceptible (either resistant or intermediate) to ciprofloxacin. Kuo et al. (26) indicated that 96% of 110 isolates recovered from diseased pigs in 2011 and 2012

**TABLE 1** The 30 most frequently isolated *Salmonella* serovars in Taiwan in 2017 to 2018 and the number of isolates for antimicrobial susceptibility testing (AST)

| Serovar | No. of isolates | % of total isolates | No. of isolates for AST | % of isolates of serovar for AST |
|---|---|---|---|---|
| Enteritidis | 2,325 | 33.9 | 804 | 34.6 |
| Typhimurium | 1,233 | 18.0 | 431 | 35.0 |
| Anatum | 921 | 13.4 | 314 | 34.1 |
| Newport/Bardo | 425 | 6.2 | 145 | 34.1 |
| Agona | 267 | 3.9 | 105 | 39.3 |
| Derby | 170 | 2.5 | 58 | 34.1 |
| Albany | 107 | 1.6 | 35 | 32.7 |
| Stanley | 106 | 1.5 | 33 | 31.1 |
| Livingstone var. 14+ | 95 | 1.4 | 32 | 33.7 |
| Weltevreden | 92 | 1.3 | 37 | 40.2 |
| Paratyphi B var. Java | 90 | 1.3 | 28 | 31.1 |
| Braenderup | 82 | 1.2 | 30 | 36.6 |
| Mbandaka | 71 | 1.0 | 25 | 35.2 |
| Goldcoast | 68 | 1.0 | 37 | 54.4 |
| Infantis | 67 | 1.0 | 35 | 52.2 |
| Virchow | 65 | 0.9 | 27 | 41.5 |
| Hadar/Istanbul | 57 | 0.8 | 20 | 35.1 |
| Brancaster | 53 | 0.8 | 21 | 39.6 |
| Montevideo | 53 | 0.8 | 23 | 43.4 |
| Bareilly | 50 | 0.7 | 21 | 42.0 |
| Potsdam | 46 | 0.7 | 16 | 34.8 |
| Give | 44 | 0.6 | 15 | 34.1 |
| Corvallis | 37 | 0.5 | 14 | 37.8 |
| Rissen | 36 | 0.5 | 13 | 36.1 |
| Schwarzengrund | 28 | 0.4 | 8 | 28.6 |
| Muenster | 21 | 0.3 | 3 | 14.3 |
| Havana | 20 | 0.3 | 8 | 40.0 |
| Itami | 20 | 0.3 | 6 | 30.0 |
| Panama | 18 | 0.3 | 10 | 55.6 |
| Kedougou | 17 | 0.2 | 5 | 29.4 |
| Other 49 serovars | 177 | 2.6 | 72 | 40.7 |
| Total | 6,861 | 100.0 | 2,431 | 35.4 |

were multidrug resistant (MDR) (resistant to three or more antimicrobial classes), 21% were ciprofloxacin resistant, and 44% were cefotaxime resistant. While resistance to fluoroquinolones and third-generation cephalosporins is increasing, azithromycin and carbapenems are considered the alternatives for the treatment of invasive salmonellosis caused by MDR and extensively drug-resistant (XDR) *Salmonella* strains (27–29). Although azithromycin resistance genes, *mph*(A) and *erm*(42), had previously been found in some MDR *S.* Typhimurium and *S.* Albany isolates from humans and animals (22, 23), the prevalence and mechanisms of azithromycin resistance in NTS in Taiwan have not been fully investigated. In this study, we investigated antimicrobial resistance and the mechanisms of azithromycin resistance in human NTS isolates from Taiwan recovered in 2017 and 2018.

## RESULTS

***Salmonella* serovars.** We collected 6,861 *Salmonella* isolates from collaborative hospitals across the country in 2017 and 2018. The isolates fell into 79 serovars; the first 10 most frequently isolated serovars were *S.* Enteritidis (33.9%), *S.* Typhimurium (18.0%), *S.* Anatum (13.4%), *S.* Newport/*S.* Bardo (6.2%), *S.* Agona (3.9%), *S.* Derby (2.5%), *S.* Albany (1.6%), *S.* Stanley (1.5%), *S.* Livingstone var. 14+ (1.4%), and *S.* Wltevreden (1.3%), which together accounted for 83.7% of the total isolates (Table 1).

**Antimicrobial susceptibility testing (AST).** Of the 6,861 isolates recovered in 2017 to 2018, 35.4% were randomly selected for antimicrobial susceptibility testing (Table 1). The susceptibility testing data indicated that 32.5% to 49.0% of the isolates were resistant

**TABLE 2** Antimicrobial resistance by percentage in nontyphoidal *Salmonella* isolates and the four most prevalent *Salmonella* serovars from Taiwan, 2017 to 2018

| Antimicrobial | Resistance rate (%) in: | | | | |
| --- | --- | --- | --- | --- | --- |
| | All (*N* = 2,431) | *S.* Enteritidis (*N* = 804) | *S.* Typhimurium (*N* = 431) | *S.* Anatum (*N* = 314) | *S.* Newport/bardo (*N* = 145) |
| Ampicillin | 49.0 | 26.9[a] | 83.8[b] | 93.9[b] | 39.3 |
| Azithromycin | 3.1 | 0.4[a] | 3.3 | 0.3[a] | 3.4 |
| AzithromycinRS | 7.2 | 2.6[a] | 7.9 | 3.5 | 9.7 |
| Cefotaxime | 20.4 | 1.9[a] | 23.2 | 93.6[b] | 9.0[a] |
| Ceftazidime | 20.3 | 1.9[a] | 22.7 | 93.9[b] | 8.3[a] |
| Chloramphenicol | 34.3 | 5.6[a] | 45.2[b] | 95.2[b] | 35.2 |
| Ciprofloxacin | 3.2 | 0.0[a] | 3.0 | 1.9 | 0.0 |
| CiprofloxacinRS | 27.5 | 11.3[a] | 13.5[a] | 93.3[b] | 15.2[a] |
| Colistin | 14.7 | 42.2[b] | 0.7[a] | 1.6[a] | 1.4[a] |
| Cotrimoxazole[a] | 32.5 | 12.9[a] | 33.6 | 93.3[b] | 23.4 |
| Ertapenem | 0.0 | 0.0 | 0.0 | 0.0 | 0.0 |
| Gentamicin | 8.1 | 1.0[a] | 23.4[b] | 1.6[a] | 9.7 |
| Nalidixic acid | 10.0 | 8.6 | 3.7[a] | 3.8[a] | 10.3 |
| Streptomycin | 36.8 | 11.3[a] | 71.9[b] | 95.9[b] | 24.1[a] |
| Sulfamethoxazole | 46.0 | 20.8[a] | 78.9[b] | 96.2[b] | 33.8[a] |
| Tetracycline | 47.2 | 17.2[a] | 79.4[b] | 95.9[b] | 51.0 |

[a]The resistance rate is significantly lower than that for all isolates (chi-square test, *P* value ≤ 0.01).
[b]The resistance rate is significantly greater than that for all isolates (chi-square test, *P* value ≤ s0.01).

to ampicillin, chloramphenicol, streptomycin, sulfamethoxazole, tetracycline, and cotrimoxazole (sulfamethoxazole-trimethoprim), and 20.3% to 20.4% were resistant to third-generation cephalosporins (ceftazidime and cefotaxime) (Table 2). Only 10.0% of isolates were resistant to nalidixic acid, and 3.2% were resistant to ciprofloxacin, but up to 27.5% were ciprofloxacin nonsusceptible. The high ciprofloxacin nonsusceptibility rate was largely contributed by *S.* Anatum, as 93.3% of *S.* Anatum isolates were ciprofloxacin nonsusceptible. Azithromycin resistance (MIC, ≥32 mg/L) was detected in 76 (3.1%) isolates, whereas nonsusceptibility (MIC, ≥16 mg/L) was detected in 174 (7.2%) isolates. Colistin resistance was detected in 14.7% of isolates, among which most were *S.* Enteritidis; 42.2% of 804 *S.* Enteritidis isolates were colistin resistant. *S.* Enteritidis could be naturally more tolerant to colistin, as 51.6% of *S.* Enteritidis isolates had a MIC near the breakpoint (2 mg/L), and 41.0% had a MIC at the breakpoint (4 mg/L) (Table 3). In contrast, only 6.2% of *S.* Enteritidis had MIC values of ≤1 mg/L, but 94.5% to 98.1% of isolates other than *S.* Enteritidis had MIC values of ≤1 mg/L. No isolates were resistant to ertapenem.

Among the four most prevalent serovars, *S.* Anatum had extremely high resistance or nonsusceptibility rates (93.3% to 96.2%) to ampicillin, cefotaxime, ceftazidime, chloramphenicol, ciprofloxacin, cotrimoxazole, streptomycin, sulfamethoxazole, and tetracycline (Table 2). *S.* Typhimurium also had high resistance rates (45.2% to 83.8%) to ampicillin, chloramphenicol, streptomycin, sulfamethoxazole, and tetracycline. In contrast, *S.* Enteritidis had the lowest resistance or nonsusceptibility rates to 12 antimicrobials, excluding colistin, nalidixic acid, and ertapenem (Table 2). *S.* Enteritidis had the highest colistin resistance rate (42.2%).

**TABLE 3** Distribution of MIC of colistin for the four most prevalent *Salmonella* serovars in Taiwan, 2017 to 2018

| Serovar | No. isolates | MIC (mg/L), % | | | | |
| --- | --- | --- | --- | --- | --- | --- |
| | | 0.5 | 1 | 2 | 4 | 8 |
| *S.* Enteritidis | 804 | 2.5 | 3.7 | 51.6 | 41.0 | 1.1 |
| *S.* Typhimurium | 431 | 84.7 | 13.5 | 1.2 | 0.7 | 0.0 |
| *S.* Anatum | 314 | 87.3 | 9.6 | 1.6 | 1.6 | 0.0 |
| *S.* Newport/Bardo | 145 | 55.9 | 38.6 | 4.1 | 0.7 | 0.7 |
| Other 75 serovars | 737 | 88.3 | 7.3 | 3.1 | 1.1 | 0.1 |

**TABLE 4** Resistance to classes of antimicrobials by percentage in nontyphoidal *Salmonella* isolates in Taiwan, 2017 to 2018

| Resistance to classes of antimicrobials | Isolates resistant to (%): | | | | | |
|---|---|---|---|---|---|---|
| | All serovars (*N* = 2,431) | *S.* Enteritidis (*N* = 804) | *S.* Typhimurium (*N* = 431) | *S.* Anatum (*N* = 314) | *S.* Newport/bardo (*N* = 145) | Other 75 serovars (*N* = 737) |
| 0 | 35.1 | 41.0 | 13.9 | 3.5 | 41.4 | 53.3 |
| 1 | 13.7 | 33.6 | 1.2 | 0.0 | 17.2 | 4.6 |
| 2 | 3.8 | 4.2 | 3.2 | 0.3 | 1.4 | 5.6 |
| 3 | 4.9 | 5.0 | 5.3 | 0.3 | 6.2 | 6.4 |
| 4 | 11.0 | 7.6 | 32.7 | 2.2 | 12.4 | 5.6 |
| 5 | 8.6 | 5.0 | 16.9 | 1.3 | 5.5 | 11.3 |
| 6 | 16.3 | 2.1 | 13.0 | 86.3 | 11.7 | 4.9 |
| 7 | 3.8 | 1.1 | 11.1 | 4.5 | 2.8 | 2.3 |
| 8 | 1.6 | 0.2 | 2.1 | 1.6 | 1.4 | 2.7 |
| 9 | 1.2 | 0.1 | 0.5 | 0.0 | 0.0 | 3.4 |
| ≥3 (MDR) | 47.3 | 21.1 | 81.7 | 96.2 | 40.0 | 36.5 |

**MDR.** Of the 2,431 isolates with AST data, 35.1% were susceptible to all antimicrobials tested and 47.3% were MDR (Table 4). Among the four most prevalent serovars, 96.2% of *S.* Anatum isolates and 81.7% of *S.* Typhimurium isolates were MDR, and only 3.5% of *S.* Anatum isolates and 13.9% of *S.* Typhimurium isolates were pan-susceptible (Table 4). In comparison with *S.* Anatum and *S.* Typhimurium, *S.* Enteritidis and *S.* Newport had much lower MDR rates, as only 21.1% of *S.* Enteritidis and 40.0% of S. Newport isolates were MDR.

**Detection of azithromycin resistance determinants.** We used the PCR method to detect azithromycin resistance determinants in 175 selected isolates, 76 of which were azithromycin resistant (MIC, ≥32 mg/L), 43 were intermediate (MIC, 16 mg/L), and 56 were susceptible (MICs, ≤8 mg/L) (Table 5). Among the 76 azithromycin-resistant isolates, 37 were detected with *mph*(A), 21 with *ramAp*, 10 with *erm*(42), 3 with *erm*(B), and 3 with *mph*(A) and *erm*(42). *ramAp* was also detected in 16 isolates with a MIC of 16 mg/L. One resistant isolate was detected with an interrupted *ramR*. Just 1 resistant, 27 intermediate, and 56 susceptible isolates had no PCR amplicon. The genetic determinants of azithromycin resistance in 53 isolates were subsequently confirmed by whole-genome sequencing (WGS).

**Azithromycin resistance determinants and MIC.** The isolates harboring *erm*(42), *erm*(B), *mph*(A), or *erm*(42)-*mph*(A) displayed higher levels of MICs (≥64 mg/L), and those harboring *ramAp* had a MIC of 16 mg/L or 32 mg/L (Table 5). The one with a defective *ramR* had a MIC of 32 mg/L. All isolates, except one, which did not carry an *erm* (42), *erm*(B), *mph*(A), *ramAp*, or defective *ramR* displayed MICs of ≤16 mg/L.

**Distribution of azithromycin resistance determinants over serovars.** The 76 azithromycin-resistant isolates belonged to 14 *Salmonella* serovars. Among the resistance

**TABLE 5** Genetic determinants for azithromycin resistance and the MIC in nontyphoidal *Salmonella* isolates

| Resistance determinant | No. of isolates with MIC (mg/L) of:[a] | | | | | | |
|---|---|---|---|---|---|---|---|
| | 4 | 8 | 16 | 32 | 64 | 128 | Total |
| *mph*(A) | | | | | 11 | 26 | 37 |
| *ramAp* | | | 16 | 21 | | | 37 |
| *erm*(42) | | | | | | 10 | 10 |
| *erm*(B) | | | | | 1 | 2 | 3 |
| *erm*(42), *mph*(A) | | | | | | 3 | 3 |
| Δ*ramR* | | | | 1 | | | 1 |
| Unknown | | | | | 1 | | 1 |
| None | 10 | 46 | 27 | | | | 83 |
| Total | 10 | 46 | 43 | 22 | 13 | 41 | 175 |

[a]Epidemiological cutoff value suggested for nontyphoidal *Salmonella*: ≥32 mg/L for resistance, ≤16 mg/L for susceptibility (57).

**TABLE 6** Distribution of resistance determinants among azithromycin-nonsusceptible nontyphoidal *Salmonella* serovars

| Serovar | No. of isolates possessing: | | | | | | | |
|---|---|---|---|---|---|---|---|---|
| | *mph*(A) | *ramAp*[a] | *erm*(42) | *erm*(B) | *erm*(42), *mph*(A) | Δ*ramR* | Unknown | Total |
| Agona | 5 | | | | | | | 5 |
| Albany | | | 10 | | | | | 10 |
| Anatum | | | | | | 1 | | 1 |
| Blockley/Haardt | 2 | | | | | | | 2 |
| Enteritidis | | | | 1 | 2 | | | 3 |
| Goldcoast | | 37 | | | | | | 37 |
| I 1,4,[5],12:i:- | 3 | | | | | | | 3 |
| London | 1 | | | | | | | 1 |
| Mbandaka | 3 | | | | | | 1 | 4 |
| Montevideo | 2 | | | | | | | 2 |
| Newport/Bardo | 5 | | | | | | | 5 |
| Thompson | 1 | | | | | | | 1 |
| Typhimurium | 9 | | | 2 | 1 | | | 12 |
| Weltevreden | 6 | | | | | | | 6 |
| Total | 37 | 37 | 10 | 3 | 3 | 1 | 1 | 92 |

[a]Of the 37 *ramAp*-carrying *S.* Goldcoast isolates, 16 exhibited a MIC of 16 mg/L and 21 exhibited a MIC of 32 mg/L.

genetic determinants, *mph*(A) was detected in 11 serovars, *ramAp* in only *S.* Goldcoast, *erm*(42) in *S.* Albany, *S.* Enteritidis, and *S.* Typhimurium, and *erm*(B) in *S.* Typhimurium and *S.* Enteritidis (Table 6). Two *S.* Enteritidis isolates and one *S.* Typhimurium isolate harbored two resistance genes, *erm*(42) and *mph*(A). The resistance in one *S.* Anatum isolate (R17.0809) was associated with a defective *ramR*, while the resistance mechanism in the *S.* Mbandaka isolate (R17.0904) was undetermined.

**WGS.** Whole-genome sequencing using the Illumina sequencing platform was performed on 81 isolates, among which 28 from 14 serovars were further sequenced using the Nanopore sequencing platform to investigate resistance genetic determinants and the vehicles for azithromycin resistance. For the isolates with Illumina sequencing data, the median genome coverage depth was 65× (28× to 125×), the median number of contigs was 117 (65 to 266), and the median $N_{50}$ value of contigs was 437,365 bp (143,948 bp to 757,431 bp). For the 28 isolates with Nanopore sequencing data, the median genome coverage depth was 263× (100× to 813×), and the median number of circular contigs was 4 (1 to 8), indicating that the isolates could harbor 0 to 7 plasmids. The sizes of chromosomes of the 28 isolates ranged from 4,645,547 bp to 5,024,703 bp (Table 7). Of the 28 chromosomes, 16 had no resistance genes detected, 2 had an IncQ replicon, and 1 had an IncC replicon.

**Vehicles for resistance determinants.** Analysis of the complete genomic sequences indicated that *erm*(42) in *S.* Albany isolate R17.5974, accompanied by *floR* and *sul2*, was carried on an integrative and conjugative element inserted in the chromosome (Table 7), which had previously been named ICE-erm42 (23). The isolate also harbored another *Salmonella* genomic island (SGI), SGI1-F, which typically carries five resistance genes, *dfrA1*, *floR*, *tet*(G), *bla*<sub>CARB-2</sub>, and *sul1* (30).

*erm*(B) in one *S.* Enteritidis and two *S.* Typhimurium isolates, accompanied by *aadA22* and *bla*$_{CMY-2}$ or *bla*$_{CMY-2}$, *sul2*, and *tet*(M), was carried on IncI1-I($\alpha$) plasmids (Table 7). The three IncI1-I($\alpha$) plasmids shared highly similar genetic structures; all carried *bla*$_{CMY-2}$ and a clustered inversion region, called shufflon (31) (Fig. 1). The two *S.* Typhimurium isolates harbored an additional large IncFIA(HI1)-IncHI1A-IncHI1B or IncC plasmid, carrying 10 and 12 resistance genes, respectively (Table 7).

Sixteen *mph*(A)-carrying isolates from 10 serovars were selected for WGS to assemble the complete genomic sequences. *mph*(A) was carried by plasmids in 13 isolates from nine serovars and by the chromosomes in three isolates from two serovars (Table 7). The *mph*(A)-carrying plasmids belonged to five incompatibility groups, including IncHI2-IncHI2A in six isolates, IncFIB(K) in three isolates, IncC in two isolates, IncHI1A-IncHI1B(pNDM-CIT) in

**TABLE 7** Antimicrobial resistance determinants and the vehicles in 28 azithromycin-resistant *Salmonella* isolates with complete genomic sequences

| Isolate ID | *Salmonella* serovar | Vehicle of resistance determinant | Size (bp) | GC content (%) | Antimicrobial resistance determinant[a] | GenBank accession no. | Reference or source |
|---|---|---|---|---|---|---|---|
| R17.5974 | Albany | Chromosome | 5,024,703 | 51.9 | *bla*$_{CARB-2}$, *dfrA1*, **erm(42)**, *floR*, *sul1*, *sul2*, *tet(G)* | CP060730 | 23 |
| R17.4111 | Enteritidis | IncI1-I(α) | 113,786 | 49.5 | *aadA22*, *bla*$_{CMY-2}$, **erm(B)** | CP063290 | This study |
| R18.1078 | Typhimurium | IncFIA(HI1)-IncHI1A-IncHI1B(R27) | 247,565 | 47.2 | *aadA1*, *aadA2*, *bla*$_{TEM-1}$, *cmlA1*, *dfrA12*, *qnrS1*, *sul2*, *sul3*, *tet(A)*, *tet(M)* | CP065568 | This study |
| R17.1451 | Typhimurium | IncI1-I(α) | 107,790 | 49.1 | *aadA22*, *bla*$_{CMY-2}$, **erm(B)** | CP065569 | This study |
|  |  | IncC | 159,330 | 52.8 | *aac(3)-IId*, *aadA2*, *aph(3'')-Ib*, *aph(3')-Ia*, *aph(6)-Id*, *bla*$_{DHA-1}$, *bla*$_{TEM-1}$, *dfrA14*, *floR*, *sul1*, *sul2*, *tet(A)* | CP063295 | This study |
| R18.1477 | Agona | IncI1-I(α) | 102,507 | 49.2 | *bla*$_{CMY-2}$, **erm(B)**, *sul2*, *tet(M)* | CP063296 | This study |
|  |  | Chromosome | 4,801,266 | 52.1 | *fosA7.2* | CP100736 | This study |
|  |  | IncHI2-IncHI2A-IncQ1 | 258,626 | 47.7 | *aac(3)-IVa*, *aadA2*, *aph(3')-Ib*, *aph(3')-IIa*, *aph(3')-Ia*, *aph(4)-Ia*, *aph(6)-Id*, *blaTEM-1*, *ble*, *dfrA12*, *floR*, **mph(A)**, *sul1*, *sul2*, *tet(A)* | CP100737 | This study |
| R18.2256 | Agona | Chromosome | 4,801,264 | 52.1 | *fosA7.2* | CP100698 | This study |
|  |  | IncHI2-IncHI2A-IncQ1 | 181,948 | 48.0 | *aac(3)-IVa*, *aadA2*, *aph(3')-Ib*, *aph(3')-IIa*, *aph(3')-Ia*, *aph(4)-Ia*, *aph(6)-Id*, *bla*$_{TEM-1}$, *ble*, *dfrA12*, **mph(A)**, *sul1*, *sul2*, *tet(A)* | CP100699 | This study |
| R17.0776 | Blockley/Haardt | Chromosome | 4,922,910 | 52.0 | *aph(3'')-Ib*, *aph(3')-Ia*, *aph(6)-Id*, **mph(A)**, *tet(A)* | CP100728 | This study |
|  |  | Inc group unidentified | 83,642 | 46.7 | *floR* | CP100729 | This study |
| R18.0186 | Blockley/Haardt | Chromosome | 4,890,321 | 52.0 | *aph(3'')-Ib*, *aph(3')-Ia*, *aph(6)-Id*, **mph(A)**, *tet(A)* | CP100710 | This study |
|  |  | Inc group unidentified | 77,896 | 46.9 | *floR* | CP100711 | This study |
| R18.1595 | London | IncFIB(K) | 113,134 | 54.0 | *aac(3)-IId*, *aac(6')-Ib-cr5*, *aadA16*, *aph(3'')-Ib*, *aph(6)-Id*, *arr-3*, *bla*$_{TEM-1}$, *catA2*, *dfrA27*, *floR*, **mph(A)**, *qnrB*, *qnrB6*, *sul1*, *sul2*, *tet(A)* | CP100694 | This study |
| R17.4855 | Mbandaka | IncHI1A-IncHI1B (pNDM-CIT) | 328,074 | 50.7 | *aadA2*, *bla*$_{TEM-1}$, *catA1*, *dfrA12*, *floR*, **mph(A)**, *sul1*, *sul2*, *tet(A)* | CP100723 | This study |
| R17.4849 | Montevideo | Col(pHAD28)-like | 84,323 | 52.5 | *aadA2*, *dfrA12*, **mph(A)**, *sul1*, *tet(B)* | CP100748 | This study |
| R18.0234 | Newport/Bardo | IncHI2-IncHI2A | 299,493 | 47.2 | *aac(3)-IVa*, *aadA1*, *aadA2*, *aadA22*, *aph(3')-IIa*, *aph(4)-Ia*, *bla*$_{TEM-135}$, *ble*, *dfrA12*, *floR*, *lnu(F)*, **mph(A)**, *qnrS1*, *sul1*, *sul3*, *tet(A)* | CP100745 | This study |
| R18.0287 | Newport/Bardo | Chromosome | 4,745,939 | 52.2 | *tet(B)* | CP100689 | This study |
|  |  | IncHI2-IncHI2A | 258,970 | 46.9 | *aac(3)-IId*, *aadA1*, *aadA2*, *aph(3'')-Ib*, *aph(3')-Ia*, *aph(6)-Id*, *bla*$_{CMY-2}$, *bleO*, *dfrA12*, *floR*, *lnu(F)*, **mph(A)**, *sul3*, *tet(A)* | CP100690 | This study |
| R18.0872 | Thompson | IncC | 149,266 | 52.9 | *aadA2*, *aph(3')-Ib*, *aph(6)-Id*, *bla*$_{CMY-2}$, *bla*$_{TEM-1}$, *dfrA12*, *floR*, **mph(A)**, *qnrS1*, *sul1*, *sul2*, *tet(A)* | CP100703 | This study |
|  |  | IncI1-I(α) | 93,173 | 49.9 | *bla*$_{CMY-2}$ | CP100704 | This study |
|  |  | Inc group unidentified | 15,860 | 49.8 | *bla*$_{CMY-2}$ | CP100706 | This study |
| R18.1932 | Typhimurium | IncC | 161,012 | 52.8 | *aac(3)-IId*, *aph(3')-Ib*, *aph(6)-Id*, *bla*$_{DHA-1}$, *bla*$_{TEM-1}$, *dfrA17*, *floR*, **mph(A)**, *qnrB4*, *sul1*, *sul2*, *tet(A)* | CP100733 | This study |
| R17.3867 | Typhimurium | Chromosome | 4,977,701 | 52.1 | *aac(3)-IVa*, *aadA2*, *aph(4)-Ia*, *bla*$_{TEM-1}$, *dfrA12*, *floR*, **mph(A)**, *sul1* | CP100691 | This study |
|  |  | IncQ1 | 11,080 | 61.8 | *aph(3'')-Ib*, *aph(6)-Id*, *sul2*, *tet(A)* | CP100692 | This study |
| R18.0409 | I 1,4,[5],12:i:- | Chromosome-IncQ1[b] | 4,962,789 | 52.2 | *aph(3'')-Ib*, *aph(6)-Id*, *bla*$_{TEM-1}$, *sul2* | CP100707 | This study |
|  |  | IncHI2-IncHI2A | 278,879 | 47.8 | *aac(3)-IVa*, *aadA1*, *aadA2*, *aadA22*, *aph(3'')-IIa*, *aph(4)-Ia*, *aph(6)-Ic*, *bla*$_{TEM-1}$, *ble*, *bleO*, *cmlA1*, *dfrA12*, *floR*, *lnu(F)*, **mph(A)**, *oqxA*, *oqxB*, *sul1*, *sul3*, *tet(A)* | CP100708 | This study |
|  |  | IncI1-I(α) | 97,209 | 49.5 | *bla*$_{CMY-2}$ | CP100709 | This study |

**TABLE 7** (Continued)

| Isolate ID | *Salmonella* serovar | Vehicle of resistance determinant | Size (bp) | GC content (%) | Antimicrobial resistance determinant[a] | GenBank accession no. | Reference or source |
|---|---|---|---|---|---|---|---|
| R17.5474 | II,4,[5],12:i:- | Chromosome-IncQ1[b] | 4,955,572 | 52.2 | *aph(3")-Ib, aph(6)-Id, bla*$_{TEM-1}$*, sul2, tet(B)* | CP100715 | This study |
| | | IncHI2-IncHI2A-IncX1 | 294,171 | 47.1 | *aac(3)-IVa, aadA1, aadA2, aph(3")-Ib, aph(3')-IIa, aph(4)-Ia, aph(6)-Id, bla*$_{DHA-1}$*, ble, cmlA1, dfrA12, floR,* **mph(A)***, qnrB4, sul1, sul3, tet(A)* | CP100716 | This study |
| R18.0830 | Weltevreden | IncFIB(K) | 78,306 | 55.6 | *aadA2, aph(3')-Ia, dfrA12,* **mph(A)***, sul1, sul2, tet(A)* | CP100697 | This study |
| R17.4942 | Weltevreden | IncFIB(K) | 71,243 | 56.5 | *aadA2, dfrA12,* **mph(A)***, sul1, sul2, tet(A)* | CP100720 | This study |
| R17.1476 | Enteritidis | Chromosome | 4,752,036 | 52.1 | *aadA1,* **erm(42)***, floR,* **mph(A)***, sul3* | CP100724 | This study |
| | | IncFIB(S)-IncFII(S) | 64,326 | 51.8 | *bla*$_{TEM-1}$ | CP100725 | This study |
| | | IncX1 | 24,484 | 44.7 | *aph(3")-Ib, aph(6)-Id, bla*$_{TEM-1}$*, sul2* | CP100726 | This study |
| R18.1630 | Enteritidis | Chromosome | 4,777,143 | 52.2 | *aadA1, aadA2, aph(3')-Ia, dfrA12,* **erm(42)***, floR,* **mph(A)***, sul1, sul3* | CP100666 | This study |
| | | IncFIB(S)-IncFII(S) | 64,325 | 51.8 | *bla*$_{TEM-1}$ | CP100667 | This study |
| | | IncX1 | 29,336 | 47.2 | *aph(3")-Ib, aph(6)-Id, bla*$_{TEM-1}$*, sul2, tet(A)* | CP100668 | This study |
| R18.0292 | Typhimurium | IncC | 199,721 | 53.4 | *aac(3)-IId, aadA2, aph(3")-Ib, aph(3')-Ia, aph(6)-Id, bla*$_{DHA-1}$*, bla*$_{TEM-1}$*, floR,* **mph(A)***, qnrB4, sul1, sul2, tet(A),* **erm(42)** | CP100740 | This study |
| R18.0450 | Goldcoast | IncI1-I(α) | 88,662 | 50.1 | **erm(42)** | CP100741 | This study |
| | | IncHI2-IncHI2A | 262,557 | 47.2 | *aac(3)-IId, aadA22, aph(3')-Ia, aph(6)-Id, bla*$_{LAP-2}$*, bla*$_{TEM-1}$*, floR, lnu(F), sul2, sul3, tet(A),* **ramAp** | CP100683 | 58 |
| R18.0877 | Goldcoast | IncHI2-IncHI2A | 278,374 | 47.3 | *aac(3)-IId, aadA22, aph(3')-Ia, aph(6)-Id, arr-2, bla*$_{CTX-M-55}$*, bla*$_{LAP-2}$*, bla*$_{TEM-1}$*, dfrA14, floR, lnu(F), qnrS13, sul2, sul3, tet(A),* **ramAp** | CP037959 | 58 |
| R18.1297 | Goldcoast | IncHI2-IncHI2A | 279,202 | 47.3 | *aac(3)-IId, aadA22, aph(3')-Ia, aph(6)-Id, arr-2, bla*$_{CTX-M-55}$*, bla*$_{LAP-2}$*, bla*$_{TEM-1}$*, dfrA14, floR, lnu(F), qnrS13, sul2, sul3, tet(A),* **ramAp** | CP100686 | This study |
| R17.0809 | Anatum | IncI2(δ) | 62,147 | 42.1 | *mcr-1.1* | CP100687 | This study |
| | | Chromosome-IncC[b] | 4,732,812 | 52.2 | *aadA2, aph(3")-Ib, aph(6)-Id, bla*$_{DHA-1}$*, dfrA23, floR, lnu(F), qnrB4, sul1, sul2, tet(A),* **ΔramR** | CP100678 | This study |
| R17.0904 | Mbandaka | IncI(γ) | 88,397 | 49.7 | *bla*$_{CMY-2}$ | CP100679 | This study |
| | | Chromosome | 4,760,001 | 52.2 | No resistance genes detected | CP100670 | This study |

[a]Azithromycin resistance determinants are indicated in bold.
[b]Chromosome inserted with a plasmid replicon of IncQ1 or IncC.

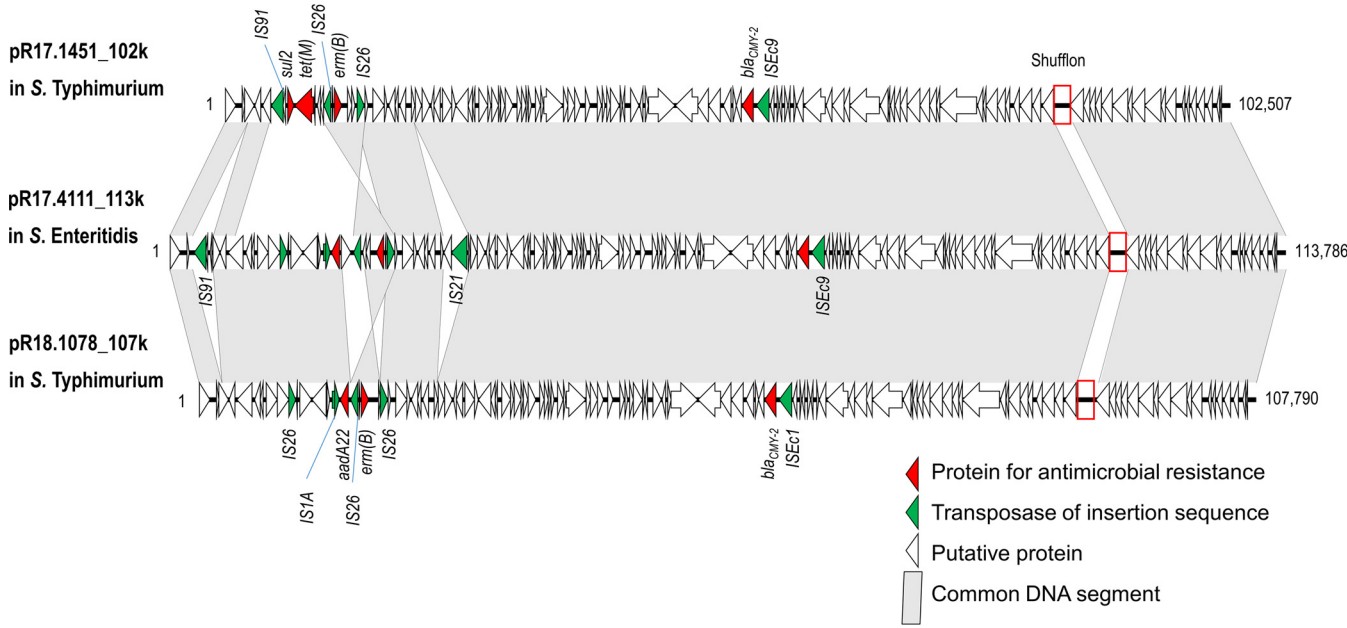

**FIG 1** Genetic maps of *erm*(B)-carrying IncI1-I(α) plasmid pR17.1451_102k in *S.* Typhimurium R17.1451 (GenBank accession no. CP063296), pR17.4111_113k in *S.* Enteritidis R17.4111 (CP063290), and plasmid pR18.1078_107k in *S.* Typhimurium R18.1078 (CP065569). Images were initially created using Easyfig v2.2.2. Open reading frames are shown as horizontal boxes, where the arrowheads indicate the direction of translation. The red rectangles mark the regions of the shufflon.

one isolate, and Col(pHAD28)-like in one isolate. IncHI2-IncHI2A plasmids in two *S.* Agona and one *S.* I 1,4,[5],12:i:- isolate harbored an additional replicon, IncQ1 and IncX1, respectively. The multiple-replicon plasmids could be derived from the fusion of IncHI2-IncHI2A and IncQ1plasmids, and IncHI2-IncHI2A and IncX1 plasmids. *mph*(A) in two *S.* Welteveden and one *S.* London isolate was carried on IncFIB(K) plasmids. A comparison of the genetic maps indicated that the two *mph*(A)-carrying IncFIB(K) plasmids in two *S.* Welteveden isolates were closely related but quite different from the one in the *S.* London isolate (Fig. 2).

*mph*(A) in two *S.* Blockley/Haardt and one *S.* Typhimurium isolate was located in the chromosomes. In the two *S.* Blockley/*S.* Haardt isolates (R17.0776 and R18.0186), *mph*(A) and four other resistance genes *aph(3′)-Ib*, *aph(3′)-Ia*, *aph(6)-Id*, and *tet*(A), were clustered in a 22,187-bp region in the chromosomes (Fig. 3). This 22,187-bp unit could be a transposable element, as it was flanked by IS*6100* and IS*26*, inserted in a gene encoding a PfkB family carbohydrate kinase, and generated an 8-bp tandem repeat at the insertion site. Similarly, *mph*(A) and seven other resistances genes, *aac(3)-IVa*, *aadA2*, *aph(4)-Ia*, *bla*TEM-1, *dfrA12*, *floR*, and *sul1*, in the *S.* Typhimurium isolate (R17.3867) were clustered in an 82,497-bp region in the chromosome (Fig. 3). This 82,497-bp genetic unit could be an IS*26* composite transposon, as it was flanked by IS*26* at both ends and generated an 8-bp tandem repeat at the insertion site.

*mph*(A) in all 16 isolates was arranged in the genetic structure of IS*26*-*mph*(A)-*mrx* (A)-*mphR*(A)-IS*6100*, which has been found in various bacterial species (32–35).

Among the three isolates carrying both *erm*(42) and *mph*(A), the resistance genes in two *S.* Enteritidis isolates (R17.1476 and R18.1630) were located in the chromosomes, whereas *erm*(42) and *mph*(A) in the *S.* Typhimurium isolate (R18.0292) were carried on an IncI1-I(α) and an IncC plasmid, respectively (Table 7). *erm*(42) and *mph*(A) in *S.* Enteritidis R17.1476, accompanying *floR*, *aadA1*, and *sul3*, were located in a 48-kb SGI flanked by IS*26* (Fig. 4), whereas *erm*(42), *mph*(A), and seven other resistance genes in *S.* Enteritidis R18.1630 were located in a 110-kb SGI flanked by IS*26*. In *S.* Enteritidis R17.1476, 95% of the sequence of the SGI was found in the SGI carried in *S.* Enteritidis R18.1630; both SGIs had five common resistance genes, but the one in R18.1630 carried four additional resistance genes, *dfrA12*, *aadA2*, *sul1*, and *aph(3′)-Ia* (Fig. 4). *mph*(A) in the chromosome of *S.* Enteritidis R17.1476 and in the IncC plasmid of *S.* Typhimurium R18.0292 was located in

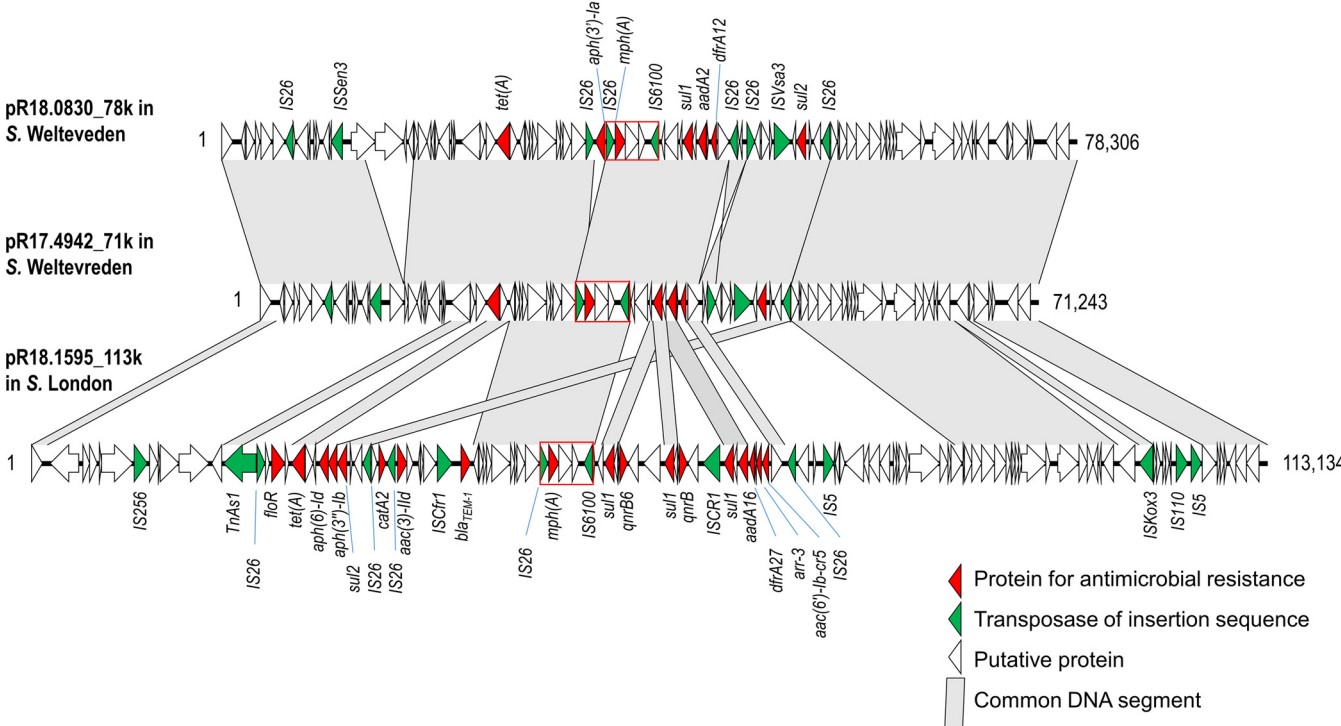

**FIG 2** Genetic maps of three *mph*(A)-carrying IncFIB(K) plasmids from *S.* Welteveden R18.0830 (GenBank accession no. CP100697), *S.* Weltevreden R17.4942 (CP100720), and *S.* London R18.1595 (CP100694). Images were initially created using Easyfig v2.2.2. Open reading frames are shown as horizontal boxes, where the arrowheads indicate the direction of translation. The red rectangles mark the IS*26-mph*(A)-*mrx*(A)-*mphR*(A)-IS*6100* unit.

the IS*26-mph*(A)-*mrx*(A)-*mphR*(A)-IS*6100* unit, whereas *mph*(A) in *S.* Enteritidis R18.1630 was located in a variant, IS*26-mph*(A)-*mrx*(A)-*mphR*(A)-IS*26*. As shown in Fig. 4, each resistance gene cluster in the SGIs was flanked by IS*26*; thus, the spread of the resistance genes could be most likely mediated by the transposition of IS*26* (36).

*ramAp*, accompanied by 11 to 15 other resistance genes, found only in *S.* Goldcoast isolates, was carried on IncHI2-IncHI2A plasmids (Table 7). The plasmid-borne *ramAp* can enhance the expression of *acrAB-tolC*, resulting in elevated MICs of many antimicrobials, including azithromycin (37).

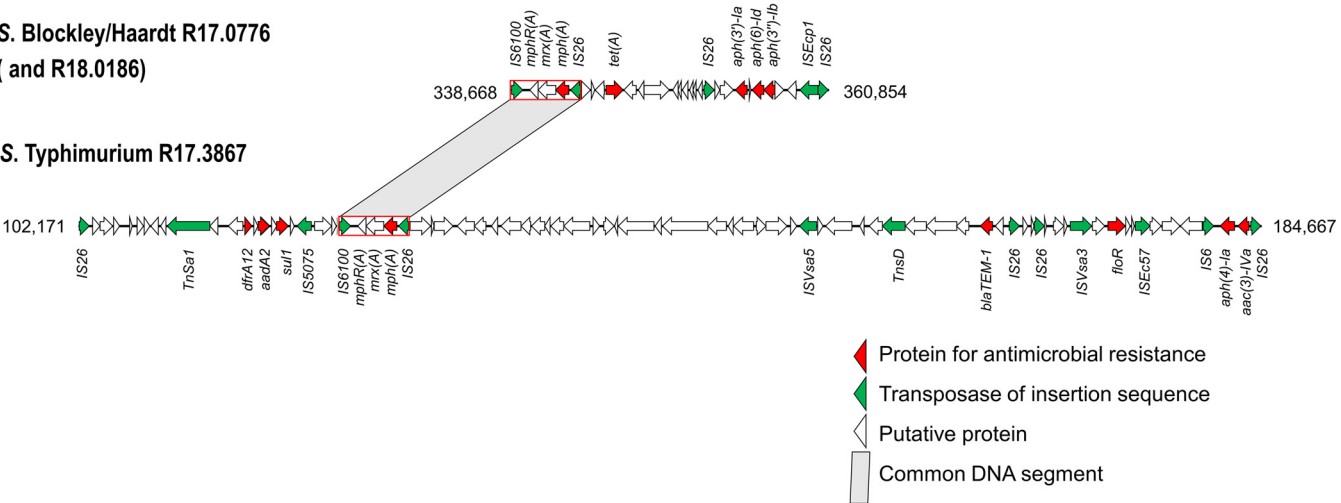

**FIG 3** Genetic maps of *mph*(A)-carrying transposable elements in the chromosomes of *S.* Blockley/Haardt R17.0776 (GenBank accession no. CP100728) and *S.* Typhimurium R17.3867 (CP100691). Images were initially created using Easyfig v2.2.2. Open reading frames are shown as horizontal boxes, where the arrowheads indicate the direction of translation. The red rectangles mark the IS*26-mph*(A)-*mrx*(A)-*mphR*(A)-IS*6100* unit.

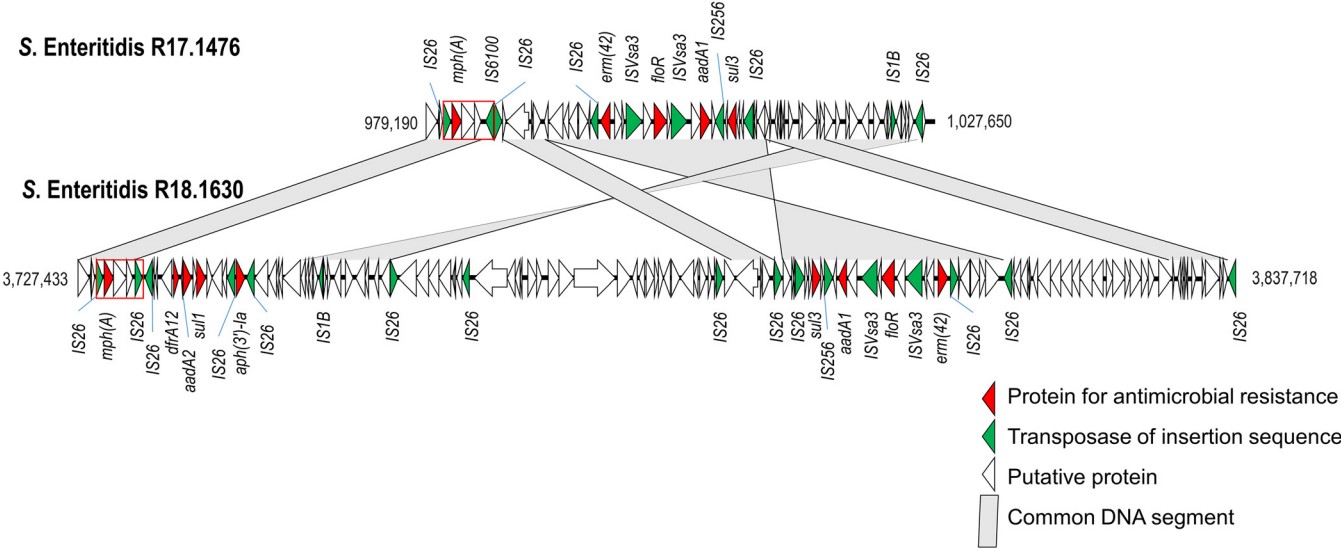

**FIG 4** Genetic maps of regions with *mph*(A), *erm*(42), and the other resistance genes in the chromosomes of *S.* Enteritidis R17.1476 (GenBank accession no. CP100724) and *S.* Enteritidis R18.1630 (CP100666). Images were initially created using Easyfig v2.2.2. Open reading frames are shown as horizontal boxes, where the arrowheads indicate the direction of translation. The red rectangles mark the IS*26*-*mph*(A)-*mrx*(A)-*mphR*(A)-IS*6100* unit in *S.* Enteritidis R17.1476 and the IS*26*-*mph*(A)-*mrx*(A)-*mphR*(A)-IS*26* unit in *S.* Enteritidis R18.1630.

Analysis of the complete genomic sequence indicated that *S.* Anatum R17.0809 had an insertional disruption in *ramR* by a 90-kb InC plasmid that carried 11 resistance genes. *ramR* is a repressor that regulates the expression of *ramA* (38). The InC plasmid was first identified in an MDR *S.* Anatum clone in Taiwan (39). In *S.* Anatum R17.0809, the whole sequence of the 90-kb IncC plasmid was inserted into *ramR* in the chromosome. Compared with isolates with *mph*(A), *erm*(B), and *erm*(42), isolates carrying a *ramAp* as well as a defective *ramR* displayed lower azithromycin MICs (16 mg/L or 32 mg/L) (Table 5).

No known genetic determinant associated with azithromycin resistance was detected in the resistant isolate, *S.* Mbandaka R17.0904.

Among the incompatibility plasmids carrying resistance genes, IncHI2-IncHI2A and IncC plasmids were more widely distributed and often carried more resistance genes than other smaller plasmids (Table 7). IncHI1A-IncHI1B(pNDM-CIT) and IncFIA(HI1)-IncHI1A-IncHI1B(R27) plasmids were the other two megaplasmids; they also carried large numbers (9 and 10) of resistance genes (Table 7).

## DISCUSSION

Our study indicates that 3.1% of the 2,431 NTS isolates in Taiwan recovered in 2017 and 2018 were resistant to azithromycin (MIC, ≥32 mg/L) (Table 2). The azithromycin resistance rate is higher than that for NTS isolates from the United States and European countries. In the United States, the azithromycin resistance rate was only 0.3% among 2,455 human NTS isolates in 2017 and 0.5% among 2,963 human NTS isolates in 2018 (data obtained from the National Antimicrobial Resistance Monitoring System [NARMS], USA; https://wwwn.cdc.gov/narmsnow/). Of the 3,467 human NTS isolates from 8 European countries in 2020, only 0.8% were resistant to azithromycin (12).

Azithromycin resistance in NTS isolates from Taiwan is very complicated. Our study indicates that azithromycin resistance has been widespread, as the resistance has been found in 14 *Salmonella* serovars (Table 6), and the resistance is mediated by multiple mechanisms, including inactivation of the drug by Mph(A), target methylation by Erm42 and ErmB, and overexpression of efflux pump(s) mediated by a plasmid-borne RamAp activator or by a detective *ramR*, a repressor gene of *ramA*.

*mph*(A), encoding a macrolide 2′ phosphotransferase, is the most frequently identified genetic determinant for azithromycin resistance and has been discovered in many bacterial species (40, 41). This study indicates that *mph*(A) is the most widely distributed gene for

azithromycin resistance in Taiwan, present in 40 azithromycin-resistant isolates from 11 *Salmonella* serovars (Table 6), and indicates that *mph*(A) is carried by various incompatibility plasmids and by chromosomes (Table 7). IncHI2-IncHI2A is the commonest replicon among the *mph*(A)-carrying plasmids. As shown in Table 7, IncHI2-IncHI2A plasmids are usually very large (181 kb to 299 kb) and can carry a large number of resistance genes. IncHI2-IncHI2A plasmids may be able to stably merge with other replicons so that they can acquire more resistance genes (42). We previously reported an *mph*(A)-carrying IncHI2-IncHI2A plasmid that was capable of moving from an *S.* Typhimurium strain to an *Escherichia coli* strain through conjugation (22). Although *mph*(A)-carrying IncHI1A-IncHI1B(pNDM-CIT) and Col (pHAD28)-like plasmids may be reported for the first time in this study, *mph*(A)-carrying IncC and IncFIB(K) plasmids may have been widespread, as a *mph*(A)-carrying IncC plasmid has been found in *Vibrio cholerae* (43), and *mph*(A)-carrying IncFIB(K) plasmid, in *Citrobacter freundii* (44). Since *mph*(A) is carried by diverse incompatibility plasmids, it would be expected to disseminate among a wide range of bacterial species.

*erm*(42) is frequently found in the major bacterial pathogens of animals, *Mannheimia haemolytica* and *Pasteurella multocida* (19, 45). In this study, *erm*(42) was found in 13 isolates, including 10 *S.* Albany, two *S.* Enteritidis, and one *S.* Typhimurium (Table 6). The resistance gene in the *S.* Albany and *S.* Enteritidis isolates was located in the chromosomes, while the gene in the *S.* Typhimurium isolate was carried by an IncI1-I($\alpha$) plasmid (Table 7). We previously found that *erm*(42) in *S.* Albany isolates was located in the chromosomes and carried by an integrative and conjugative element (ICE) called ICE_erm42 (23). ICE_erm42 was able to move from *S.* Albany strains to distantly related bacterial species, *Vibrio cholerae* (23). ICE_erm42-carrying *S.* Albany strains were first identified in Taiwan in 2014 and have become prevalent in the country since then (23). *erm*(42) has also been found to be carried on other ICEs in *Pasteurella multocida* and *Actinobacillus pleuropneumoniae* (46, 47). As *erm*(42) was carried in mobile plasmids and ICEs, this resistance determinant could soon be disseminated among diverse bacterial species.

*erm*(B) is considered to originate from Gram-positive bacteria because of the low GC (G+C) content, but it has already been found in many Gram-negative bacterial species (48). *erm*(B) has frequently been reported in Gram-negative *Campylobacter* spp., an organism having a low GC content of around 30.5% (49, 50). Recently, our laboratory identified *erm*(B) in 30.0% of *Campylobacter coli* isolates and 1.7% of *Campylobacter jejuni* isolates from humans with campylobacteriosis in Taiwan from 2016 to 2019 (51). In this study, we found that *erm*(B) was carried by IncI1-I($\alpha$) plasmids in two *S.* Enteritidis and one *S.* Typhimurium isolate (Table 7). The *erm*(B) sequences from the three isolates have around 33.3% GC content, while *Salmonella* has a GC content of around 52% (Table 7). The three *erm*(B)-carrying IncI1-I($\alpha$) plasmids have highly similar genetic structures (Fig. 1), suggesting that they could be highly mobile and may have been introduced into the two *Salmonella* serovars recently.

Efflux pumps play a role in macrolide resistance (13, 52). The AcrAB-TolC efflux pump, which belongs to the resistance-nodulation-division (RND) transporter family, can pump out a very wide range of drugs, including macrolides (53). Overexpression of AcrAB and TolC or substitutional mutation at the AcrB codon 717 can result in increased azithromycin MICs (21). The expression of the AcrAB-TolC efflux pump can be facilitated by the transcriptional activator RamA, while the expression of *ramA* is negatively regulated by the RamR transcriptional repressor (54). Thus, overexpression of RamA activator or loss of RamR repressor can lead to increased expression of the AcrAB-TolC efflux pump and extrusion of more antimicrobials (55). We previously identified a plasmid-borne *ramA*, designated as *ramAp*, in XDR *S.* Goldcoast strains (37). The plasmid-borne *ramAp* has an identical amino acid sequence to the *ramA* in the chromosome of *Klebsiella quasipneumoniae* and 92% (104/113) sequence identity with the *ramA* in the chromosomes of *Salmonella* strains (37). *ramAp* has the same function as *ramA*; it can lead to increased expression of AcrAB-TolC and elevated MICs of 2- to 4-fold to chloramphenicol, azithromycin, nalidixic acid, ciprofloxacin, sulfamethoxazole, trimethoprim, tetracycline, and tigecycline (37). However, whether the expression of *ramAp* is regulated by the

transcriptional repressor RamR remains to be investigated. Our study indicates that the 21 azithromycin-resistant *S.* Goldcoast isolates harbor a plasmid-borne *ramAp* but no other known azithromycin resistance determinants; thus, the azithromycin resistance in the *S.* Goldcoast isolates is likely to be mediated by RamAp, whereas the resistance in the *S.* Anatum isolate is associated with the defective RamR repressor that leads to over-expression of RamA and increased azithromycin MICs.

The *S.* Goldcoast isolates with a *ramAp* displayed azithromycin MICs of 16 mg/L or 32 mg/L (Table 5). Clinical and Laboratory Standards Institute (CLSI) breakpoints for azithromycin are only established for *S.* Typhi, with a MIC of ≤16 mg/L as susceptible and ≥32 mg/L as resistant (56). The azithromycin interpretive standards for NTS serovars have not been established by CLSI. An epidemiological cutoff value of ≤16 mg/L of azithromycin has been proposed for wild-type *Salmonella* (57); however, this criterion could include *ramAp*-carrying strains that display an azithromycin MIC of 16 mg/L. Thus, the epidemiological cutoff value of azithromycin for NTS should be reconsidered. In this study, we consider isolates with an azithromycin MIC of ≤8 mg/L to be susceptible, 16 mg/L to be intermediate, and ≥32 mg/L to be resistant.

We are not able to identify the mechanism of azithromycin resistance in *S.* Mbandaka R17.0904 by identifying relevant resistance genetic determinants from the complete genome sequence. We did not find resistance genes and mutations associated with azithromycin resistance in 23S RNA, ribosomal proteins L4 and L22, and AcrB. However, we measured an 8-fold-decreased MIC in a test with efflux pump inhibitor phenylalanine-arginine β-naphthylamide (data not shown). The enhanced capacity of efflux pump(s) could have played a major role in resistance in the isolate.

Our data indicate that NTS isolates from Taiwan recovered in 2017 and 2018 have high resistance rates to the conventional first-line drugs, ampicillin, chloramphenicol, and cotrimoxazole. In comparison with the resistance rates for the NTS isolates recovered in 1998 to 2002, the resistance rates for the three drugs are about the same or even lower (25). However, the resistance rates for third-generation cephalosporins (cefotaxime and ceftazidime) increased from almost zero for the isolates from 1998 to 2002 to 20% for the isolates recovered in 2017 and 2018. The increasing antimicrobial resistance in NTS in Taiwan is a great public health issue that needs to be taken seriously. Recently, we have witnessed the emergence and dissemination of MDR *S.* Anatum and XDR *S.* Goldcoast in Taiwan (39, 58). The XDR *S.* Goldcoast strains harbor a plasmid-borne *ramAp* and 15 resistance genes and are nonsusceptible to azithromycin but susceptible to carbapenems (37). MDR and XDR NTS strains may easily acquire genes for azithromycin and carbapenem resistance, resulting in the emergence of more resistant strains that could lead to difficulty in the medical treatment of invasive infections. Recently, we identified carbapenem-resistant XDR *S.* Goldcoast strains from a salmonellosis outbreak that occurred at a hospital in Taiwan; the carbapenem resistance in the XDR strains was due to the acquisition of a $bla_{OXA-48}$-carrying plasmid (59).

In conclusion, NTS isolates from Taiwan are highly resistant, as nearly half of NTS isolates are MDR. The azithromycin resistance rate (3.1%) for NTS isolates from Taiwan is much higher than the resistance rates for the NTS isolates from European countries and the United States. The azithromycin resistance is primarily mediated by *mph*(A), *erm*(B), *erm*(42), and *ramAp*. As the resistance determinants are primarily carried by mobile genetic elements, including plasmids and transposable elements, they could easily be transferred among human bacterial pathogens. Further surveillance and research are needed for monitoring the epidemiological trend of resistance and the dissemination of the azithromycin resistance determinants among human bacterial pathogens.

## MATERIALS AND METHODS

**Bacterial isolates.** *Salmonella* isolates, recovered from salmonellosis patients, were obtained from 30 collaborative hospitals across Taiwan in 2017 and 2018. The collection of isolates was approved by the Institutional Review Board of the Taiwan Centers for Disease Control (Taiwan CDC), and the institutional review board waived the need for informed consent (IRB107111). The isolates were subjected to pulsed-field gel electrophoresis (PFGE) genotyping using the standardized PulseNet PFGE protocol (60),

and the serotypes of isolates were determined through PFGE pattern comparison with those in the *Salmonella* database established by Taiwan CDC (61).

**Antimicrobial susceptibility testing.** *Salmonella* isolates were tested for susceptibility to 14 antimicrobials using the broth microdilution method with a custom-made 96-well Sensititre MIC panel (TREK Diagnostic Systems Ltd., West Essex, England) and performed according to the manufacturer's instructions. The MIC breakpoints for *Enterobacterales* set by the Clinical and Laboratory Standards Institute (CLSI) were used to interpret the AST results of ampicillin, cefotaxime, ceftazidime, chloramphenicol, ciprofloxacin, colistin, cotrimoxazole (trimethoprim-sulfamethoxazole), gentamicin, nalidixic acid, sulfamethoxazole, and tetracycline (56). For azithromycin, a MIC of ≥32 mg/L and a MIC of ≤16 mg/L were set to be resistant and susceptible, respectively (57); however, in this study, a MIC of 16 mg/L azithromycin was interpreted to be intermediate. For streptomycin, MICs of ≥64 mg/L, 32 mg/L, and ≤16 mg/L were set to be resistant, intermediate, and susceptible, respectively.

**Detection of genetic determinants relevant to azithromycin resistance.** PCR was performed on 175 isolates to detect the genetic determinants associated with azithromycin resistance. The panel of isolates included all 76 isolates with an azithromycin MIC of ≥32 mg/L, 43 isolates with a MIC of 16 mg/L, and 56 isolates with a MIC of ≤8 mg/L (Table 5). The targets of PCR detection included *erm*(42) (primers 5′-TGCACCAT CTTACAAGGAGT and 5′-CATGCCTGTCTTCAAGGTTT) (62), *erm*(B) (5′-GAAAAGGTACTCAACCAAATA and 5′-AG TAACGGTACTTAAATTGTTTAC) (63), *mph*(A) (5′-GTGAGGAGGAGCTTCGCGAG and 5′-GATACCTCCCAACTGTA CGCA) (63), *ramAp* (5′-ACGATTTCCGCTCAGGTGAT and 5′-CGGGTAAAGGTCTGTTGCGA) (37), and *ramR* (5′-CGT GTCGATAACCTGAGCGG and 5′-AAGGCAGTTCCAGCGCAAAG) (64).

**Whole-genome sequencing and analysis.** WGS of bacterial isolates was conducted in the Central Region Laboratory of Taiwan CDC using the Illumina MiSeq (https://www.illumina.com) and the Nanopore MinION (https://nanoporetech.com/) sequencing platforms. DNA of isolates was extracted using the Qiagen DNeasy blood and tissue kit (Qiagen Co., Germany). For Illumina sequencing, the library construction was performed using the Illumina DNA prep (M) tagmentation system (Illumina Co., USA), and sequencing was run with the MiSeq reagent kit v3 (2 × 300 cycles) following the manufacturer's instructions. For Nanopore sequencing, genomic DNA was treated with the reagents of the rapid barcoding kit to generate barcoded sequencing libraries and then was run through a MinION flow cell. The raw signal data (fast5) were converted into nucleotide sequences (FASTQ) using the base calling tool Guppy (Oxford Nanopore Technologies). The FASTQ sequences were assembled with Illumina reads using the Unicycler v0.4.8 (65) to obtain complete genomic sequences. The assembled sequences were subsequently polished using POLCA v4.0.5 (66) and Polypolish v0.5.0 (67). The polished complete genomic sequences were subjected to the identification of antimicrobial resistance genes and plasmid incompatibility groups using the tools ResFinder v4.1.11 and PlasmidFinder v2.1.6, provided by the Center for Genomic Epidemiology of the Technical University of Denmark (http://www.genomicepidemiology.org/). Insertion sequences were identified using the tool of ISfinder, which is accessible at the website https://www-is.biotoul.fr/. The integrative and conjugative element ICE_erm42 was identified and the mobility of the element was proven in a previous study (23). For Fig. 1 to 4, genes (open-reading frames) were annotated using RAST (https://rast.nmpdr.org/) (68), and BLAST comparisons between multiple genomic regions were performed using a genome comparison visualizer, Easyfig v2.2.2 (https://mjsull.github.io/Easyfig/) (69).

**Data availability.** All assembled complete sequences of plasmids and the chromosomes for 28 *Salmonella* isolates have been deposited in the database of the National Center for Biotechnology Information under the accession numbers listed in Table 7. The complete genomic sequences of *S.* Albany R17.5974 and *S.* Goldcoast R18.0877 were assembled and submitted to the NCBI in previous studies (23, 58).

## ACKNOWLEDGMENTS

This study was funded by the Ministry of Health and Welfare, Taiwan (grant number MOHW109-CDC-C-315-144406).

We sincerely thank the hospitals for providing *Salmonella* isolates for disease surveillance. The hospitals include Changhua Christian Hospital, Chiayi Chang Gung Memorial Hospital, Chiayi Christian Hospital, China Medical University Hospital, Chung Shan Medical University Hospital, Far Eastern Memorial Hospital, Hsin Chu Armed Force Hospital, Hualien Tzu Chi Hospital, Kaohsiung Chang Gung Memorial Hospital, Kaohsiung Medical University Chung-Ho Memorial Hospital, Kaohsiung Veterans General Hospital, Linkou Chang Gung Memorial Hospital, Lotung Pohai Hospital, MacKay Memorial Hospital, Mackay Memorial Hospital Hsinchu Branch, Military Taoyuan General Hospital, National Cheng Kung University Hospital, National Taiwan University Hospital, National Taiwan University Hospital Yun-Lin Branch, Pingtung Christian Hospital, Show Chwan Memorial Hospital, Shuangho Hospital of Ministry of Health and Welfare, Taichung Armed Forces General Hospital, Taichung Veterans General Hospital, Taipei Tzu Chi Hospital, Taipei Veterans General Hospital, Taiwan Landseed Hospital, Ton Yen General Hospital, Tri-Service General Hospital, and Wei Gong Memorial Hospital.

We declare that we have no known competing financial interests or personal relationships that have or could be perceived to have influenced the work reported in this article.

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
