## [Reviewer comments · Microbiology Spectrum]

Microbiology Spectrum

Antimicrobial resistance and mechanisms of azithromycin resistance in nontyphoidal *Salmonella* isolates in Taiwan, 2017–2018

Chien-Shun Chiou, Yu-Ping Hong, You-Wun Wang, Bo-Han Chen, Ru-Hsiou Teng, Hui-Yung Song, and Ying-Shu Liao

Corresponding Author(s): Chien-Shun Chiou, Taiwan Centers for Disease Control

Review Timeline:

Submission Date:	August 24, 2022
Editorial Decision:	November 14, 2022
Revision Received:	November 30, 2022
Editorial Decision:	December 22, 2022
Revision Received:	December 28, 2022
Accepted:	January 4, 2023

Editor: Cheryl Andam

Reviewer(s): Disclosure of reviewer identity is with reference to reviewer comments included in decision letter(s). The following individuals involved in review of your submission have agreed to reveal their identity: Samuel J. Bloomfield (Reviewer #1)

Transaction Report:

DOI: <https://doi.org/10.1128/spectrum.03364-22>

November 14, 2022

Dr. Chien-Shun Chiou
Centers for Disease Control
Centers for Diagnostics and Vaccine Development
5F 20 Wen-Sin South 3rd Road
Taichung 40855
Taiwan

Re: Spectrum03364-22 (Antimicrobial resistance and mechanisms of azithromycin resistance in nontyphoidal Salmonella isolates in Taiwan, 2017–2018)

Dear Dr. Chien-Shun Chiou:

Link Not Available

Sincerely,

Cheryl Andam

Journals Department
Reviewer comments:

Reviewer #1 (Comments for the Author):

Antimicrobial resistance and mechanisms of azithromycin resistance in nontyphoidal Salmonella isolates in Taiwan, 2017–2018 by Chiou et al. clearly demonstrates the problem with azithromycin resistance in NTS from Taiwan, the AMR mechanisms associated with this resistance and what mobile genetic elements are associated with the AMR mechanisms. The analysis of the NTS isolates and their genomes is thorough, but a few more details would benefit the manuscript. In addition there are a few times in the manuscript where the authors could be more specific. More details on these can be found below. Overall, the manuscript is well designed and highlights the need for surveillance of azithromycin in Taiwan and worldwide.

Major comments

Lines 16-37: I think some of the methods section needs to be added to the abstract, specifically the phenotypic, PCR and WGS methods and how many isolates were analysed using these methods.

Lines 417-434: There is nothing wrong with the approach taken to sequence and assemble the genomes. However, incomplete and misassemblies can still occur. It would be good to have a paragraph in the results section describing the quality of the assemblies, e.g., the number of contigs, how many contigs were circular, were any plasmid replicons found in chromosomes and what are the ribosomal chromosome arrangements (these can easily be determined using Socru). Also were there any discrepancies between the PCR and whole genome sequencing AMR gene results

Minor comments

Line 23: Instead of using the term "over" I would give the exact percentages.

Lines 58-61: It would be useful to distinguish which serovars cause typhoid fever and which cause paratyphoid fever.

Line 134-139. I understand that you are attempting to categorise the antimicrobials based on how many Salmonella are resistant, but I would still give the exact percentages for each antimicrobial agent.

Line 140. I am assuming by nonsusceptible you mean that they are either resistant or intermediate, but it might be good to define this the first time you use the term.

Line 151-155: For this paragraph, I would use actual percentages of isolates belonging to each serovar rather than using the term "most". In addition, I would not use the term "significantly" unless a statistical model was used.

Line 197: Instead of saying around a dozen I would give the range of resistance genes found on these plasmids.

Lines 288-293: It might be best to cite additional papers that have found these plasmids in other species.

Line 390-393: For the isolates that were whole genome sequenced, were the serovars predicted using PFGE the same as those predicted using whole genome sequencing software, e.g. SISTR.

Lines 428-434: Please provide the versions of the bioinformatics software used.

Line 429-434. How were insertion sequences and transposons (including ICEs) identified?

Figure 2-5. What software was used to form these alignments and how were the non-AMR gene annotated? Also, many of the alignments have multiple copies of IS26, is this significant?

Grammatical errors

A few grammatical errors were identified. I have done my best to suggest correction for these, but I may have missed some.

Lines 24-26: This sentence is a little confusing. I would change "mph(A), ramAp, erm(42), erm(B), defective ramR, and probably the enhanced expression of efflux pump(s)" to "mph(A), erm(42), erm(B), and possibly the enhanced expression of efflux pump(s) due to ramAp or defective ramR".

Lines 66-67: I would replace the term "first cause" with "largest cause".

Lines 74-75: I would replace "As the widespread resistance to the conventional first-line drugs among Salmonella serovars" to "Due to the widespread resistance of Salmonella serovars to conventional first-line drugs".

Lines 87-88 I would replace "is inevitable to drive the development of resistance in bacteria" to "selects for resistance to this antimicrobial in bacteria".

Line 89: I would replace "the resistance" with "resistance".

Line 91: I would replace "Macrolides express their antibacterial activity" to "Macrolides inhibit bacteria".

Line 99: I would remove "an".

Line 270: I would remove the "as many as".

Line 272: "Mph(A)" should not be italicised.

Line 295, 297, 298, 305 and 314: I would replace "is" with "was".

Line 301: I would replace "could move" to "has previously been shown to be able".

Line 318 and 355: I would replace "should" with "could".

Line 319: I would remove "quite".

Line 328: I would replace "thus facilitating the extrusion capacity of" to "extrusion of more".

Line 372: I would replaced "developed through" to "due to".

Line 423: Missing closing bracket.

Reviewer #3 (Comments for the Author):

Antimicrobial resistance and mechanisms of azithromycin resistance in nontyphoidal Salmonella isolates in Taiwan, 2017–2018; Chiou et al

General comments

Chiou et al have presented a detailed overview of AMR in NTS Salmonellae in recent years in Taiwan. They speculate on mechanisms of Azithromycin resistance. The manuscript is detailed however in that detail, the novelty and impact of the study is lost. As azithromycin resistance was only reported in ~3% of isolates, I'm unsure why they focused on this. While the significance is stated in the "importance" section, it is not made very clear throughout the manuscript. There was no effort to

describe novel resistance mechanisms, especially in the isolate that was phenotypically resistant but didn't encode any known AMR determinants. The discussion around changing the breakpoints for different serovars is good. Additional subheadings in the results would also help improve the flow of that section.

Minor comments:

1. The introduction and discussion are very long. Please consider shortening it by removing information or condensing. Some sentences can be condensed, for example, instead of "in a previous study, we found", say "we previously showed". There are many cases of this and similar examples.
2. Please clarify the significance of azithromycin resistance throughout the manuscript.
3. Please highlight the novel aspects of your work.
 - a. In particular, please clarify which plasmids were novel, and which ones are already published.
4. A heatmap or phylogenetic tree summarising the presence of AMR determinants across the serovars would be a really nice way to convey some of the data rather than relying on large tables.
5. A figure correlating the major AMR determinants with MIC values, and any discrepancies, would greatly enhance the paper.
 - a. Performing statistical analysis on these would further strengthen the paper.
6. Please state in the intro (eg line 91) that azithromycin is a macrolide.
7. Through the discussion, "should" and "would" are used when "could" or "may" are more appropriate - please amend these.
8. Some statements are not very scientific, for example, "are more/less resistant" (eg line 115, 151, 161 and others) needs to be clarified and/or quantified - are you referring to MIC values, the number of AMR genes, or the prevalence of AMR isolates?
9. Line 188-192 - so two copies of floR? Are these paralogs? Are they encoded on the same element?
10. Line 197 - around a dozen - be specific - give the range
11. Line 231 - "relatively different"... ? be specific - how much sequence identity or number of genes shared?

Tables & figures:

1. In table 2, I don't think it is accurate to run statistics grouping all other serovars for comparison against the most prevalent ones. Please remove.
2. In table 3, please state that these are percentages, or alternatively, provide the raw values.
 - a. Also, I suggest this table should be supplementary, as these results are implied in table 2.
3. In table 4, please note the MIC breakpoints
4. In table 6, please replace vehicle with compartment
 - a. Also, I suggest this table should be supplementary, and instead to include a summary as a main table
 - b. Further, why are some of the cells in the "Antimicrobial resistance determinant" column blank?
5. Figure 1 should not be a line graph - please replace with a table.
6. Figures 2-5:
 - a. please include a key for the gene colour
 - b. please include a key for the grey shading
 - c. please clarify which are novel sequences and which are previously published
 - d. the "sufflon" was not mentioned in the text - please explain this
 - e. please label boundaries of MGEs such as the ICEs mentioned in the text.

Minor English changes:

1. Line 39 - "major public health concern"
2. Line 146 - instinctively > naturally
3. "in recent" > recently

Staff Comments:

Preparing Revision Guidelines

Please return the manuscript within 60 days; if you cannot complete the modification within this time period, please contact me. If you do not wish to modify the manuscript and prefer to submit it to another journal, please notify me of your decision immediately so that the manuscript may be formally withdrawn from consideration by Microbiology Spectrum.

Antimicrobial resistance and mechanisms of azithromycin resistance in nontyphoidal *Salmonella* isolates in Taiwan, 2017–2018; Chiou et al

General comments

Chiou et al have presented a detailed overview of AMR in NTS *Salmonellae* in recent years in Taiwan. They speculate on mechanisms of Azithromycin resistance. The manuscript is detailed however in that detail, the novelty and impact of the study is lost. As azithromycin resistance was only reported in ~3% of isolates, I'm unsure why they focused on this. While the significance is stated in the "importance" section, it is not made very clear throughout the manuscript. There was no effort to describe novel resistance mechanisms, especially in the isolate that was phenotypically resistant but didn't encode any known AMR determinants. The discussion around changing the breakpoints for different serovars is good. Additional subheadings in the results would also help improve the flow of that section.

Minor comments:

1. The introduction and discussion are very long. Please consider shortening it by removing information or condensing. Some sentences can be condensed, for example, instead of "in a previous study, we found", say "we previously showed". There are many cases of this and similar examples.
2. Please clarify the significance of azithromycin resistance throughout the manuscript.
3. Please highlight the novel aspects of your work.
 - a. In particular, please clarify which plasmids were novel, and which ones are already published.
4. A heatmap or phylogenetic tree summarising the presence of AMR determinants across the serovars would be a really nice way to convey some of the data rather than relying on large tables.
5. A figure correlating the major AMR determinants with MIC values, and any discrepancies, would greatly enhance the paper.
 - a. Performing statistical analysis on these would further strengthen the paper.
6. Please state in the intro (eg line 91) that azithromycin is a macrolide.
7. Through the discussion, "should" and "would" are used when "could" or "may" are more appropriate – please amend these.
8. Some statements are not very scientific, for example, "are more/less resistant" (eg line 115, 151, 161 and others) needs to be clarified and/or quantified – are you referring to MIC values, the number of AMR genes, or the prevalence of AMR isolates?
9. Line 188-192 – so two copies of floR? Are these paralogs? Are they encoded on the same element?
10. Line 197 – around a dozen – be specific – give the range
11. Line 231 – "relatively different" ... ? be specific – how much sequence identity or number of genes shared?

Tables & figures:

1. In table 2, I don't think it is accurate to run statistics grouping all other serovars for comparison against the most prevalent ones. Please remove.
2. In table 3, please state that these are percentages, or alternatively, provide the raw values.
 - a. Also, I suggest this table should be supplementary, as these results are implied in table 2.
3. In table 4, please note the MIC breakpoints
4. In table 6, please replace vehicle with compartment
 - a. Also, I suggest this table should be supplementary, and instead to include a summary as a main table
 - b. Further, why are some of the cells in the "Antimicrobial resistance determinant" column blank?
5. Figure 1 should not be a line graph – please replace with a table.
6. Figures 2-5:
 - a. please include a key for the gene colour
 - b. please include a key for the grey shading
 - c. please clarify which are novel sequences and which are previously published
 - d. the "sufflon" was not mentioned in the text – please explain this
 - e. please label boundaries of MGEs such as the ICEs mentioned in the text.

Minor English changes:

1. Line 39 – "major public health concern"
2. Line 146 – instinctively > naturally
3. "in recent" > recently

Reviewer #1

1. Overall, the manuscript is well-designed and highlights the need for surveillance of azithromycin in Taiwan and worldwide.

Reply: Thanks.

2. Lines 16-37: I think some of the methods section needs to be added to the abstract, specifically the phenotypic, PCR, and WGS methods and how many isolates were analysed using these methods.

Reply: The methods (AST, PCR, WGS) and the number of isolates tested have been added in the abstract, to be “Antimicrobial resistance was investigated in 2,341 nontyphoidal *Salmonella* (NTS) isolates recovered from humans in Taiwan between 2017 and 2018 using antimicrobial susceptibility testing. Azithromycin resistance determinants were detected in 175 isolates using polymerase-chain-reaction and confirmed in 81 isolates using whole-genome sequencing” (Lines 16–20 in the revised manuscript).

3. Lines 417-434: There is nothing wrong with the approach taken to sequence and assemble the genomes. However, incomplete and misassemblies can still occur. It would be good to have a paragraph in the results section describing the quality of the assemblies, e.g., the number of contigs, how many contigs were circular, were any plasmid replicons found in chromosomes, and what are the ribosomal chromosome arrangements (these can easily be determined using Socru).

Reply: A paragraph has been added in the result section to describe the quality of NGS data, to be “WGS. Whole-genome sequencing using the Illumina sequencing platform was performed on 81 isolates among which 28 from 14 serovars were further sequenced using the Nanopore sequencing platform to investigate resistance genetic determinants and the vehicles for azithromycin resistance. For the isolates with Illumina sequencing data, the median genome coverage depth was 65x (28–125x), the median number of contigs was 117 (65–266), and the median N50 of contigs was 437,365 bp (143,948–757,431 bp). For the 28 isolates with Nanopore sequencing data, the median genome coverage depth was 263x (100–813x), and the median number of circular contigs was 4 (1–8), indicating that the isolates could harbor 0 to 7 plasmids. The sizes of chromosomes of the 28 isolates ranged from 4,645,547 bp to 5,024,703 bp (Table 7). Of the 28 chromosomes, 16 had no resistance genes

detected, 2 had an IncQ replicon, and 1 had an IncC replicon” (Lines 192–203).

Since we want to focus on antimicrobial resistance and the mechanisms of azithromycin resistance, we think it is better not to present the ribosomal chromosome arrangements in the 28 *Salmonella* isolates in this manuscript.

4. Also were there any discrepancies between the PCR and whole genome sequencing AMR gene results?

Reply: The azithromycin resistance genes detected by the PCR are all concordantly identified in the whole genome sequences.

5. Line 23: Instead of using the term "over" I would give the exact percentages.

Reply: The exact percentages have been added to describe the resistance rates, to be “Multidrug resistance was found in 47.3% of all isolates and 96.2% of *S. Anatum* and 81.7% of *S. Typhimurium* isolates. Resistance to the conventional first-line drugs (ampicillin, chloramphenicol, and cotrimoxazole), extended-spectrum cephalosporins (cefotaxime and ceftazidime), and ciprofloxacin was found in 32.5–49.0%, 20.3–20.4%, and 3.2% of isolates, respectively” (Lines 20–24).

6. Lines 58-61: It would be useful to distinguish which serovars cause typhoid fever and which cause paratyphoid fever.

Reply: The serovars that cause typhoid and paratyphoid fever have been indicated, to be “Typhoidal *Salmonella* serovar, *S. Typhi*, and paratyphoidal serovars, *S. Paratyphi A*, *S. Paratyphi B*, *S. Paratyphi C*, and *S. Sendai*, can cause invasive systemic infections in humans and higher primates...” (Lines 63–65).

7. Line 134-139. I understand that you are attempting to categorise the antimicrobials based on how many *Salmonella* are resistant, but I would still give the exact percentages for each antimicrobial agent.

Reply: The description has been revised, to be “Of the 6,861 isolates recovered in 2017–2018, 35.4% were randomly selected for antimicrobial susceptibility testing (Table 1). The susceptibility testing data indicated that 32.5%–49.0% of the isolates were resistant to ampicillin, chloramphenicol, streptomycin, sulfamethoxazole, tetracycline, and

cotrimoxazole (sulfamethoxazole-trimethoprim), and 20.3%–20.4% were resistant to third-generation cephalosporins (ceftazidime and cefotaxime) (Table 2)” (Lines 135–140).

8. Line 140. I am assuming by nonsusceptible you mean that they are either resistant or intermediate, but it might be good to define this the first time you use the term.

Reply: The term “nonsusceptible” has been defined when it appears for the first time, to be “nonsusceptible (either resistant or intermediate)” (Line 115).

9. Line 151-155: For this paragraph, I would use actual percentages of isolates belonging to each serovar rather than using the term "most". In addition, I would not use the term "significantly" unless a statistical model was used.

Reply: Thanks for the suggestion. We have revised the paragraph by giving the actual percentages, to be “Among the four most prevalent serovars, *S. Anatum* had extremely high resistance or nonsusceptibility rates (93.3% to 96.2%) to ampicillin, cefotaxime, ceftazidime, chloramphenicol, ciprofloxacin, cotrimoxazole, streptomycin, sulfamethoxazole, and tetracycline (Table 2). *S. Typhimurium* also had high resistance rates (45.2% to 83.8%) to ampicillin, chloramphenicol, streptomycin, sulfamethoxazole, and tetracycline. In contrast, *S. Enteritidis* had the lowest resistance or nonsusceptibility rates to 12 antimicrobials, excluding azithromycin, colistin, nalidixic acid, and ertapenem (Table 2). *S. Enteritidis* had the highest colistin resistance rate (42.2%)” (Lines 153–160).

10. Line 197: Instead of saying around a dozen I would give the range of resistance genes found on these plasmids.

Reply: The exact number of resistance genes has been given, to be “...carrying 10 and 12 resistance genes, respectively (Table 7)” (Line 215).

11. Lines 288-293: It might be best to cite additional papers that have found these plasmids in other species.

Reply: mph(A)-carrying plasmids found in bacterial species other than *Salmonella* have been searched and described in the context. Papers are

cited, to be “Although *mph(A)*-carrying IncHI1A-IncHI1B(pNDM-CIT) and Col(pHAD28)-like plasmids could be reported for the first time in this study, *mph(A)*-carrying IncC and IncFIB(K) plasmids could have been widespread as *mph(A)*-carrying IncC plasmid has been found in *Vibrio cholerae* (43) and *mph(A)*-carrying IncFIB(K) plasmid in *Citrobacter freundii* (44)” (Lines 304–308).

12. Line 390-393: For the isolates that were whole genome sequenced, were the serovars predicted using PFGE the same as those predicted using whole genome sequencing software, e.g. SISTR.

Reply: Yes, the serovars determined via PFGE patterns comparison with those in a *Salmonella* PFGE database are completely concordant with the serovars predicted from WGS data using the SISTR and SeqSero.

13. Lines 428-434: Please provide the versions of the bioinformatics software used.

Reply: The versions of the bioinformatics software have been indicated, to be “ The fastq sequences were assembled with Illumina reads using the Unicycler v 0.4.8 (65) to obtain complete genomic sequences. The assembled sequences were subsequently polished using POLCA v 4.0.5 (66) and Polypolish v 0.5.0 (67). The polished complete genomic sequences were subjected to the identification of antimicrobial resistance genes and plasmid incompatibility groups using the tools of ResFinder v 4.1.11 and PlasmidFinder v 2.1.6 provided by the Center for Genomic Epidemiology of the Technical University of Denmark (<http://www.genomicepidemiology.org/>)” (Lines 443–450).

14. Line 429-434. How were insertion sequences and transposons (including ICEs) identified?

Reply: Identification of insertion sequences and the ICE_erm42 has been described in the revised manuscript, to be “Insertion sequences were identified using the tool of ISfinder, which is accessible at the website: <https://www-is.biotoul.fr/>. The integrative and conjugative element, ICE_erm42 was identified and the mobility of the element was proven in a previous study (23)” (Lines 450–453).

15. Figure 2-5. What software was used to form these alignments and how were the non-AMR gene annotated? Also, many of the alignments have

multiple copies of IS26, is this significant?

Reply: The software used to make figures 1–4 has been described, as “For figures 1–4, genes (open-reading frames) were annotated using RAST (<https://rast.nmpdr.org/>) (68) and BLAST comparisons between multiple genomic regions were performed using a genome comparison visualizer, Easyfig v2.2.2 (<https://mjsull.github.io/Easyfig/>) (69)” (Lines 453–456)

IS26 is very abundant in Enterobacterales, it has been proven to be instrumental in the rearrangement and spread of multiple antibiotic resistance (a detail review: Varani et al. 2021. The IS6 family, a clinically important group of insertion sequences including IS26. *Mob DNA* 12:11).

16. Lines 24-26: This sentence is a little confusing. I would change "mph(A), ramAp, erm(42), erm(B), defective ramR, and probably the enhanced expression of efflux pump(s)" to "mph(A), erm(42), erm(B), and possibly the enhanced expression of efflux pump(s) due to ramAp or defective ramR"

Reply: Thanks for the suggestion. The sentence has been revised according to the suggestion.

17. Lines 66-67: I would replace the term "first cause" with "largest cause".
Reply: Thanks for the correction. It has been revised according to the suggestion.

18. Lines 74-75: I would replace "As the widespread resistance to the conventional first-line drugs among Salmonella serovars" to "Due to the widespread resistance of Salmonella serovars to conventional first-line drugs"

Reply: Thanks for the suggestion. The sentence has been revised according to the suggestion.

19. Lines 87-88 I would replace "is inevitable to drive the development of resistance in bacteria" to "selects for resistance to this antimicrobial in bacteria"

Reply: Thanks for the suggestion. The sentence has been revised according to the suggestion.

20. Line 89: I would replace "the resistance" with "resistance"

Reply: Thanks for the suggestion. It has been revised.

21. Line 91: I would replace "Macrolides express their antibacterial activity" to "Macrolides inhibit bacteria"

Reply: Thanks for the suggestion. It has been revised according to the suggestion.

22. Line 99: I would remove "an"

Reply: Thanks. The word has been removed.

23. Line 270: I would remove the "as many as"

Reply: Thanks. The words have been removed from the sentence.

24. Line 272: "Mph(A)" should not be italicised.

Reply: Thanks. The word has been modified.

25. Line 295, 297, 298, 305 and 314: I would replace "is" with "was".

Reply: Thanks. The tense of the verbs has been changed to past time.

26. Line 301: I would replace "could move" to "has previously been shown to be able"

Reply: Thanks for the suggestion. The sentence has been revised according to the suggestion.

27. Line 318 and 355: I would replace "should" with "could"

Reply: Thanks. The word has been replaced.

28. Line 319: I would remove "quite"

Reply: Thanks. It has been removed.

29. Line 328: I would replace "thus facilitating the extrusion capacity of "to "extrusion of more"

Reply: Thanks for the suggestion. The sentence has been revised according to the suggestion.

30. Line 372: I would replaced "developed through" to "due to"

Reply: Thanks. The sentence has been revised.

31. Line 423: Missing closing bracket.

Reply: Thanks for pointing out the error.

Reviewer #3

32. The introduction and discussion are very long. Please consider shortening it by removing information or condensing. Some sentences can be condensed, for example, instead of "in a previous study, we found", say "we previously showed". There are many cases of this and similar examples.

Reply: Thanks for the comments. We have revised the wording as possible and removing some information from paragraphs.

33. Please clarify the significance of azithromycin resistance throughout the manuscript.

Reply: Thanks for the suggestion. We have addressed the significance of azithromycin resistance in the abstract section as "Azithromycin and carbapenems are the last resort for the treatment of invasive salmonellosis caused by multidrug-resistant (MDR) and extensively drug-resistant *Salmonella* strains" (lines 41–43), and the introduction section as "While resistance to fluoroquinolones and third-generation cephalosporins is increasing, azithromycin and carbapenems are considered the alternatives for the treatment of invasive salmonellosis caused by MDR and extensively drug-resistant (XDR) *Salmonella* strains (27-29)" (lines 118–121).

34. Please highlight the novel aspects of your work.

a. In particular, please clarify which plasmids were novel, and which ones are already published.

Reply: Thanks for the suggestion. We have emphasized one of the important findings in the Importance section, to be "Our study also indicates that azithromycin resistance is primarily mediated by *mph(A)*, *erm(42)*, *erm(B)*, and *ramAp*, which are mostly carried on mobile genetic elements. Although the *mph(A)*-carrying IncHI1A-IncHI1B(pNDM-CIT) and Col(pHAD28)-like plasmids could be reported for the first time in this study, *mph(A)*-carrying IncHI2-IncHI2A, IncC, and IncFIB(K) plasmids have frequently been found in various species of *Enterobacteriale* and even in *Vibrio cholerae*" (Lines 48–54).

35. A heatmap or phylogenetic tree summarising the presence of AMR

determinants across the serovars would be a really nice way to convey some of the data rather than relying on large tables.

Reply: Thanks for the suggestion. In this study, 81 isolates from 18 serotypes were subjected to whole-genome sequencing, with an average of only 4.5 (1–20) isolates per serovar sequenced. Thus, the data could not be sufficient to demonstrate a good relationship between AMR determinants and serovars. Nevertheless, we are preparing a manuscript that describes the distribution of AMR determinants and plasmids from a large number (558) of *S. Typhimurium* genomes.

36. A figure correlating the major AMR determinants with MIC values, and any discrepancies, would greatly enhance the paper.

a. Performing statistical analysis on these would further strengthen the paper.

Reply: Thanks for the suggestion. Because AMR determinants were identified from only 81 isolates, the amount of data may not be sufficient to conduct this type of analysis. However, this is a good point; we may conduct such an analysis in the next study. We have more than 2,000 *Salmonella* isolates with WGS and MIC data.

37. Please state in the intro (eg line 91) that azithromycin is a macrolide.

Reply: Thanks for the suggestion. We have revised the sentence to be “Macrolides, such as azithromycin and erythromycin, inhibit bacteria by...” (Line 97).

38. Through the discussion, "should" and "would" are used when "could" or "may" are more appropriate - please amend these.

Reply: Thanks for the suggestion. We have checked the whole manuscript and revised most of the verbs.

39. Some statements are not very scientific, for example, "are more/less resistant" (eg line 115, 151, 161 and others) needs to be clarified and/or quantified - are you referring to MIC values, the number of AMR genes, or the prevalence of AMR isolates?

Reply: Thanks for the comments. We have revised such kind of statement by using the exact numbers or percentages.

40. Line 188-192 - so two copies of *floR*? Are these paralogs? Are they

encoded on the same element?

Reply: Yes, the strain harbors two copies of *floR* in two different genomic islands; one is named SG11 and the other is an ICE (ICE_erm42).

41. Line 197 - around a dozen - be specific - give the range

Reply: Thanks. The sentence has been revised to be “The two *S. Typhimurium* isolates harbored an additional large IncFIA(HI1)-IncHI1A-IncHI1B or IncC plasmid, carrying 10 and 12 resistance genes, respectively (Table 7)” (Lines 213–215).

42. Line 231 - "relatively different"... ? be specific - how much sequence identity or number of genes shared?

Reply: Thanks for the comments. We have revised the whole paragraph to be “Among the 3 isolates carrying both *erm(42)* and *mph(A)*, the resistance genes in 2 *S. Enteritidis* isolates (R17.1476 and R18.1630) were located in the chromosomes, whereas *erm(42)* and *mph(A)* in the *S. Typhimurium* isolate (R18.0292) were carried on an IncI1-I(α) and an IncC plasmid, respectively (Table 7). *erm(42)* and *mph(A)* in *S. Enteritidis* R17.1476, accompanying with *floR*, *aadA1*, and *sul3*, were located in a 48-kb genomic island flanked by IS26 (Fig. 4). Whereas *erm(42)*, *mph(A)*, and 7 other resistance genes in *S. Enteritidis* R18.1630 were located in an 110-kb genomic island flanked by IS26. In *S. Enteritidis* R17.1476, 95% of the sequence of genomic island was found in the genomic island in *S. Enteritidis* R18.1630; both genomic islands had 5 common resistance genes but the one in R18.1630 carried 4 additional resistance genes *dfra12*, *aadA2*, *sul1*, and *aph(3')-Ia* (Fig. 4). *mph(A)* in the chromosome of *S. Enteritidis* R17.1476 and the IncC plasmid of *S. Typhimurium* R18.0292 was located in IS26-*mph(A)*-*mrx(A)*-*mphR(A)*-IS6100 unit, whereas *mph(A)* in *S. Enteritidis* R18.1630 was located in a variant, IS26-*mph(A)*-*mrx(A)*-*mphR(A)*-IS26. As shown in Fig. 4, each resistance gene cluster in the genomic islands was flanked by IS26, thus the spread of the resistance genes could be most likely mediated by the transposition of IS26” (Lines 242–257).

43. In table 2, I don't think it is accurate to run statistics grouping all other serovars for comparison against the most prevalent ones. Please remove.

Reply: Thanks for the comment. The column “all other serovars” has been removed from the table.

44. In table 3, please state that these are percentages, or alternatively, provide the raw values.
- a. Also, I suggest this table should be supplementary, as these results are implied in table 2.
- Reply: Thanks for the suggestion. We have stated that the data are in percentages in the Table caption. Since this is a small table, we think it is better to put it in the text rather than in a supplementary file.
45. In table 4, please note the MIC breakpoints
- Reply: We have added a footnote in Table 4 to give the ECV value, to be “Epidemiological cutoff value suggested for nontyphoidal *Salmonella*: ≥ 32 mg/L for resistance, ≤ 16 mg/L for susceptibility”
46. In table 6, please replace vehicle with compartment
- a. Also, I suggest this table should be supplementary, and instead to include a summary as a main table
- b. Further, why are some of the cells in the "Antimicrobial resistance determinant" column blank?
- Reply: We have replaced the column heading to be “Vehicle of resistance determinant” We think it is better to put this table in the text to allow readers to find the data easily.
- The blank cells have filled with “No resistance genes detected”
47. Figure 1 should not be a line graph - please replace with a table.
- Reply: Thanks. We have presented the data in Figure 1 by a table (Table 3).
48. Figures 2-5:
- a. please include a key for the gene colour
- b. please include a key for the grey shading
- c. please clarify which are novel sequences and which are previously published
- d. the "sufflon" was not mentioned in the text - please explain this
- e. please label boundaries of MGEs such as the ICEs mentioned in the text.
- Reply: We have added a figure indicator in each of Figures 1–4 (original figures 2–5) to indicate gene color and grey shading.

All the sequences in Figures 1–4 are presented for the first time. Figure 1 presents the sequence alignment of 3 *erm(B)*-carrying IncI1-I(α) plasmids, Figure 2 presents the sequence alignment of 3 *mph(A)*-carrying IncFIB(K) plasmids, Figure 3 shows the sequence alignment of 2 *mph(A)*-carrying transposable elements, and Figure 4 compares the 2 regions of genomic islands with *mph(A)* and *erm(42)*. Among the 28 isolates with complete genome sequences assembled, *S. Albany* R17.5974 (harboring an ICE_erm42) and *S. Goldcoast* R18.0877 (harboring a IncHI2-IncHI2A plasmid with a ramAp) had previously been published; we have stated in the Data Availability, to be “The complete genomic sequences of *S. Albany* R17.5974 and *S. Goldcoast* R18.0877 were assembled and submitted to the NCBI database in previous studies”

The “shufflon” has been described briefly in the Result section, to be “The 3 IncI1-I(α) plasmids shared highly similar genetic structures, all carried *bla*_{CMY-2} and a clustered inversion region, called shufflon (31)” (Lines 212–213).

We think that only the sequences in Figure 3 could be called IS26 composite transposons because we found tandem repeats of 8 bp at two sides of the IS26 at the ends. IS26 typically generates an 8-bp tandem repeat at the insertion site, as stated “This 22,187-bp unit could be a transposable element as it was flanked by IS6100 and IS26, inserted in a gene encoding a PfkB family carbohydrate kinase, and generated an 8-bp tandem repeat at the insertion site. Similarly, *mph(A)* and 7 other resistances genes *aac(3)-IVa*, *aadA2*, *aph(4)-Ia*, *bla*_{TEM-1}, *dfrA12*, *floR*, and *sul1*, in the *S. Typhimurium* isolate (R17.3867) were clustered in an 82,497-bp region in the chromosome (Fig. 3). This 82,497-bp genetic unit could be an IS26 composite transposon as it was flanked by IS26 at both ends and generated an 8-bp tandem repeat at the insertion site” (Lines 232–239).

49. Line 39 - "major public health concern"

Reply: Thanks for the correction of the usage.

50. Line 146 - instinctively > naturally

Reply: Thanks for the correction of the word usage.

51. "in recent" > recently

Reply: Thanks for the correction of the word usage.

December 22, 2022

Dr. Chien-Shun Chiou
Taiwan Centers for Disease Control
Centers for Diagnostics and Vaccine Development
5F 20 Wen-Sin South 3rd Road
Taichung 40855
Taiwan

Re: Spectrum03364-22R1 (Antimicrobial resistance and mechanisms of azithromycin resistance in nontyphoidal Salmonella isolates in Taiwan, 2017–2018)

Dear Dr. Chien-Shun Chiou:

Link Not Available

Sincerely,

Cheryl Andam

Journals Department
Reviewer comments:

Reviewer #1 (Comments for the Author):

The authors have made sufficient corrections to the comments I made and present a manuscript that clearly outlines the azithromycin resistance mechanisms used by non-typhoidal Salmonella in Taiwan and how they are transferred

Reviewer #3 (Comments for the Author):

General comments:

The paper reads a lot more clearly and the addition of the keys to the figure help the reader follow them. In future I'd suggest for sequence comparison figures to include the closest published sequence as some form of reference to anchor your comparisons. Some changes are needed to table 7.

Minor comments:

1. There is some clarification needed around ramAp:

a. the importance section states "...ramAp, which are mostly carried on mobile genetic elements" - are these genes most frequently carried on MGEs or are most of the genes listed carried on MGEs?

b. The abstract states "ramAp was a plasmid-borne ramA"; however this is not mentioned anywhere in the results, but there are several references to the two versions of RamA in the discussion. Can you please note whether ramA and ramAp have the same function, and how genetically similar they are? And does ramR have the same effect on ramAp as it does on ramA (repressor activity).

c. Can you explain or speculate as to why you didn't detect any chromosomal ramA genes?

2. You should abbreviate genomic island to GI after its first use.

3. You should abbreviate mobile genetic element to MGE after its first use.

4. Change numbers less than 10 to the word eg 5 > five

5. Line 335 - please change "should" to "may"

6. Line 335 - please change "should" to "is likely to"

7. Table 2 - it should be noted why the results for only 4 serovars are presented (as per table 3)

8. Table 7

a. please note the reason for the bolded genes (I'm guessing azithro resistance determinants?! - see comment e)

b. vehicle of resistance determinate should be two columns - vehicle would be chromosome or plasmid and the second column would be plasmid Inc type

c. I'm sorry but I still don't understand why there are rows with chromosome information without resistance genes. Please remove. Accessions for all data should be noted separately.

d. What is "Chromosome-IncQ1"? This is not mentioned in the text. If it is, it is not clear.

i. Further, it is listed in the mphA row but there is no mphA gene on that element, if I am reading the table correctly. Can you please clarify this?

ii. There are other examples eg two rows below - Inc11-I(α) only carrying bla_{CMY-2}. I suggest to rectify this confusion you could remove the third column (Azithromycin resistance determinant), as you duplicate this information in the second last column anyway.

Staff Comments:

Preparing Revision Guidelines

Please return the manuscript within 60 days; if you cannot complete the modification within this time period, please contact me. If you do not wish to modify the manuscript and prefer to submit it to another journal, please notify me of your decision immediately so that the manuscript may be formally withdrawn from consideration by Microbiology Spectrum.

Antimicrobial resistance and mechanisms of azithromycin resistance in nontyphoidal *Salmonella* isolates in Taiwan, 2017–2018, Chiou et al

General comments:

The paper reads a lot more clearly and the addition of the keys to the figure help the reader follow them. In future I'd suggest for sequence comparison figures to include the closest published sequence as some form of reference to anchor your comparisons. Some changes are needed to table 7.

Minor comments:

1. There is some clarification needed around ramAp:
 - a. the importance section states "...ramAp, which are mostly carried on mobile genetic elements" – are these genes most frequently carried on MGEs or are most of the genes listed carried on MGEs?
 - b. The abstract states "ramAp was a plasmid-borne ramA"; however this is not mentioned anywhere in the results, but there are several references to the two versions of RamA in the discussion. Can you please note whether ramA and ramAp have the same function, and how genetically similar they are? And does ramR have the same effect on ramAp as it does on ramA (repressor activity).
 - c. Can you explain or speculate as to why you didn't detect any chromosomal ramA genes?
2. You should abbreviate genomic island to GI after its first use.
3. You should abbreviate mobile genetic element to MGE after its first use.
4. Change numbers less than 10 to the word eg 5 > five
5. Line 335 – please change "should" to "may"
6. Line 335 – please change "should" to "is likely to"
7. Table 2 – it should be noted why the results for only 4 serovars are presented (as per table 3)
8. Table 7
 - a. please note the reason for the bolded genes (I'm guessing azithro resistance determinants?! – see comment e)
 - b. vehicle of resistance determinate should be two columns – vehicle would be chromosome or plasmid and the second column would be plasmid Inc type
 - c. I'm sorry but I still don't understand why there are rows with chromosome information without resistance genes. Please remove. Accessions for all data should be noted separately.
 - d. What is "Chromosome-IncQ1"? This is not mentioned in the text. If it is, it is not clear.
 - i. Further, it is listed in the mphA row but there is no mphA gene on that element, if I am reading the table correctly. Can you please clarify this?
 - ii. There are other examples eg two rows below - Inc11-I(α) only carrying bla_{cmY-2}. I suggest to rectify this confusion you could remove the third column (Azithromycin resistance determinant), as you duplicate this information in the second last column anyway

Reviewer #3

1. The importance section states "...*ramAp*, which are mostly carried on mobile genetic elements" - are these genes most frequently carried on MGEs or are most of the genes listed carried on MGEs?
Reply: Thanks for pointing out this imprecise description. The sentence has been revised to be "Our study also indicates that azithromycin resistance is primarily mediated by *mph(A)*, *erm(42)*, *erm(B)*, and *ramAp*, which are frequently carried on mobile genetic elements"
2. The abstract states "*ramAp* was a plasmid-borne *ramA*"; however this is not mentioned anywhere in the results, but there are several references to the two versions of *RamA* in the discussion. Can you please note whether *ramA* and *ramAp* have the same function, and how genetically similar they are? And does *ramR* have the same effect on *ramAp* as it does on *ramA* (repressor activity).
Reply: We discuss *ramAp* as "The plasmid-borne *ramAp* has an identical amino acid sequence to the *ramA* in the chromosome of *Klebsiella quasipneumoniae* and 92% (104/113) sequence identity with the *ramA* in the chromosomes of *Salmonella* strains (37). *ramAp* has the same function as *ramA*, it can lead to increased expression of AcrAB-TolC and elevated MICs of 2–4 folds to chloramphenicol, azithromycin, nalidixic acid, ciprofloxacin, sulfamethoxazole, trimethoprim, tetracycline, and tigecycline (37). However, whether the expression of *ramAp* is regulated by the transcriptional repressor *RamR* is remained to be investigated"
3. Can you explain or speculate as to why you didn't detect any chromosomal *ramA* genes?
Reply: *ramA* may exist in all *Enterobacteriales*. We did not show *ramA* as it was present in all the chromosomes.
4. You should abbreviate genomic island to GI after its first use.
Reply: Thanks for the suggestion. We have replaced *Salmonella* genomic islands with SGI after its first use.
5. You should abbreviate mobile genetic element to MGE after its first use.
Reply: Since "mobile genetic elements" appear only three times (in Abstract, Importance, and Conclusion), we feel it is better not to

abbreviate it.

6. Change numbers less than 10 to the word eg 5 > five
Reply: Thanks. All the numbers less than 10 have been revised.
7. Line 335 - please change "should" to "may"
Reply: Thanks. The verb has been changed according to the suggestion.
8. Line 352 - please change "should" to "is likely to"
Reply: Thanks. The verb has been changed according to the suggestion.
9. Table 2 - it should be noted why the results for only 4 serovars are presented (as per table 3)
Reply: We have modified the table caption to “Antimicrobial resistance by percentage in nontyphoidal *Salmonella* isolates and the four most prevalent *Salmonella* serovars from Taiwan, 2017-2018”
10. Table 7: please note the reason for the bolded genes (I'm guessing azithro resistance determinants?!)
Reply: A footnote has been given to denote the bolded genes.
11. Table 7: vehicle of resistance determinate should be two columns - vehicle would be chromosome or plasmid and the second column would be plasmid Inc type
Reply: We keep one column for the vehicle of resistance determinants. Since we have given a footnote to explain “chromosome-IncQ1 and chromosome-IncC”, readers can understand that the incompatibility replicons mean plasmids.
12. Table 7: I'm sorry but I still don't understand why there are rows with chromosome information without resistance genes. Please remove. Accessions for all data should be noted separately.
Reply: Thanks. We have removed the chromosomes with no resistance genes from Table 7. We agree that the revision is better to fit what the table caption indicates.
13. Table 7: What is "Chromosome-IncQ1"? This is not mentioned in the text. If it is, it is not clear.

Reply: We have given a footnote to explain what are chromosome-IncQ1 and chromosome-IncC.

14. Table 7: Further, it is listed in the mphA row but there is no mphA gene on that element, if I am reading the table correctly. Can you please clarify this?

Reply: Thanks for the comment. We have removed the column for azithromycin resistance determinant from Table 7.

15. Table 7: There are other examples eg two rows below - IncI1-I(α) only carrying bla_{cmY-2}. I suggest rectifying this confusion you could remove the third column (Azithromycin resistance determinant), as you duplicate this information in the second last column anyway.

Reply: Thanks. We have removed the column "Azithromycin resistance determinant" from Table 7.

January 4, 2023

Dr. Chien-Shun Chiou
Taiwan Centers for Disease Control
Centers for Diagnostics and Vaccine Development
5F 20 Wen-Sin South 3rd Road
Taichung 40855
Taiwan

Re: Spectrum03364-22R2 (Antimicrobial resistance and mechanisms of azithromycin resistance in nontyphoidal *Salmonella* isolates in Taiwan, 2017–2018)

Dear Dr. Chien-Shun Chiou:

Your manuscript has been accepted, and I am forwarding it to the ASM Journals Department for publication. You will be notified when your proofs are ready to be viewed.

Sincerely,

Cheryl Andam
Editor, Microbiology Spectrum
